# Normalization and effective learning rates in reinforcement learning

**Clare Lyle**[†]   **Zeyu Zheng**[†]   **Khimya Khetarpal**[†]   **James Martens**[†]   **Hado van Hasselt**[†]

**Razvan Pascanu**[†]                    **Will Dabney**[†]

## Abstract

Layer normalization has demonstrated remarkable effectiveness at preventing plasticity loss in continual and reinforcement learning (RL), though the precise reasons for this effectiveness remain mysterious. In this work, we identify new mechanisms by which layer normalization can help – and hinder – training in neural networks, and leverage these insights to improve the robustness of gradient-based optimization algorithms to nonstationarity. Our analysis reveals a surprising ability of layer normalization to revive dormant ReLU units, along with an under-appreciated vulnerability to unconstrained decay of the effective learning rate (ELR), which can drive loss of plasticity in long-running nonstationary tasks. Motivated by these findings, we propose Normalize-and-Project (NaP), a simple protocol designed to provide the numerous benefits of normalization while ensuring that the effective learning rate remains constant throughout training. To do so, NaP couples the insertion of normalization layers with weight projection. This technique mitigates loss of plasticity in two challenging continual learning problems: a sequential supervised learning task, and a continual variant of the Arcade Learning Environment. Further, by using NaP to explicitly control the effective learning rate in deep RL agents, we find that in fact the implicit ELR decay induced by parameter norm growth in these agents is critical to their ability to achieve competitive performance, suggesting the common practice of using constant learning rates in deep RL may be far from optimal.

## 1   Introduction

Many of the most promising application areas of deep learning, in particular deep reinforcement learning (RL), require training on a problem which is in some way nonstationary. In order to perform well on a nonstationary problem, the neural network must maintain its ability to adapt to new information, i.e. it must remain *plastic*. Several recent works have shown that loss of plasticity can present a major barrier to performance improvement in RL and in continual learning [Dohare et al., 2021, Lyle et al., 2021, Nikishin et al., 2022]. These works have proposed a variety of explanations for plasticity loss such as the accumulation of saturated ReLU units and increased sharpness of the loss landscape [Lyle et al., 2023], along with mitigation strategies, such as resetting dead units [Sokar et al., 2023] and regularizing the parameters towards their initial values [Kumar et al., 2023]. Many of these explanations and their corresponding mitigation strategies center around reducing drift in the distribution of pre-activations [Lyle et al., 2024], a problem which has historically been resolved in the supervised learning setting, and more recently in continual learning and RL [Hussing et al., 2024, Ball et al., 2023], by incorporating normalization layers into the network architecture.

---

[†]Google DeepMind. Correspondence to `clarelyle@google.com`

38th Conference on Neural Information Processing Systems (NeurIPS 2024).

While effective, normalization on its own is insufficient to avoid loss of plasticity [Lee et al., 2023]. Part of the reason for this lies in a subtle property of normalization: a normalization layer causes the subnetwork preceding it to become scale-invariant, which means that the layer's *effective learning rate* (ELR) now depends on the norm of its parameters [Van Laarhoven, 2017]. In particular, when the norm of the parameters grows, as it typically does in neural networks trained without regularization, the effective learning rate shrinks. Weight decay can address this problem, but runs the risk of either failing to fully mitigate the norm growth or over-regularizing the model to the point of slowing down learning and thus requires careful tuning, adding additional complexity to the training protocol.

The aim of this work is to provide a principled strategy for the use of layer normalization in non-stationary learning problems. To do so, we first identify two critical properties of layer normalization: its ability to facilitate the revival of dormant neurons, and its vulnerability to vanishing gradients as the parameter norm grows. In particular, we show that layer normalization buffers against dormant neurons due to its effect of mixing gradient signal between units. This analysis motivates Normalize-and-Project (NaP), a method which inserts normalization layers prior to each nonlinearity in a network architecture and maintains constant per-layer parameter norm. A network trained with the NaP protocol stands to benefit from the increased robustness to unit saturation provided by LayerNorm's gradient-mixing property, while also avoiding its vulnerability to vanishing gradients under growing parameter norms. Indeed, we observe empirically that NaP virtually eliminates loss of plasticity in multiple challenging non-stationary learning problems, including sequential training on games from the Arcade Learning Environment (ALE) [Bellemare et al., 2013, Abbas et al., 2023]. We further confirm that NaP can be seamlessly integrated into standard computer vision and sequence modeling baselines from standard benchmarks without hindering performance.

We further leverage Normalize-and-Project as an analytical tool to explain the dramatic performance degradations induced by weight decay in deep RL agents. By making the effective learning rate explicit, NaP reveals that the implicit learning rate decay induced by parameter norm growth in Rainbow [Hessel et al., 2018] agents is in fact critical to their performance in the ALE benchmark: certain components of the value function require a sufficiently small ELR in order to be learned, and an optimization process which does not reach this value will therefore underfit the value function in ways that can inhibit performance improvement. Indeed, while we demonstrate that the implicit schedule induced by parameter norm growth outperforms a constant effective learning rate, an explicit schedule with more aggressive learning rate decay outperforms the implicit one on the ALE benchmark. These findings are at odds the common folk wisdom that the continual nature of RL makes it unsuitable for learning rate decay, and demonstrate the untapped potential of step size schedules to accelerate deep reinforcement learning.

## 2 Background and related work

We begin by providing background on trainability and its loss in nonstationary learning problems. We additionally give an overview of neural network training dynamics and effective learning rates.

### 2.1 Training dynamics and plasticity in neural networks

Early work on neural network initialization centered around the idea of controlling the norm of the activation vectors [LeCun et al., 2002, Glorot and Bengio, 2010, He et al., 2015] using informal arguments. More recently, this perspective has been formalized and expanded [Poole et al., 2016, Daniely et al., 2016, Martens et al., 2021] to include the inner-products between pairs of activation vectors (for different inputs to the network). The function that describes the evolution of these inner-products determines the network's gradients at initialization up to rotation, and this in turn determines trainability (which was shown formally in the Neural Tangent Kernel regime by Xiao et al. [2020] and Martens et al. [2021]). A variety of initialization methods have been developed to ensure the network avoids "shattering" [Balduzzi et al., 2017] or collapsing gradients [Poole et al., 2016, Martens et al., 2021, Zhang et al., 2021b].

Once training begins, learning dynamics can be well-characterized in the infinite-width limit by the neural tangent kernel and related quantities [Jacot et al., 2018, Yang, 2019], although in practice optimization dynamics diverge significantly from this limit [Fort et al., 2020]. A number of beneficial phenomena emerge in the finite-width, finite-step-size regime, such as the self-stabilization of gradient descent [Lewkowycz et al., 2020, Cohen et al., 2021, Agarwala et al., 2022] and implicit

regularization [Barrett and Dherin, 2020, Smith et al., 2020] as a result of the non-linear training dynamics. However, particularly in non-stationary learning problems, neural networks can also be vulnerable to *loss of plasticity* [Sodhani et al., 2020, Dohare et al., 2021, Nikishin et al., 2022, Abbas et al., 2023, Lyle et al., 2024] as they drift away from the initial parameters. This phenomenon has been shown to present a limiting factor to performance in a number of RL tasks [Igl et al., 2021, Lyle et al., 2021, Nikishin et al., 2023], along with continual learning and warm-starting neural network training [Berariu et al., 2021, Ash and Adams, 2020].

A variety of architectural choices can accelerate the training of extremely deep networks, including residual connections [He et al., 2016] and normalization layers [Ioffe and Szegedy, 2015, Ba et al., 2016]; these methods have also been demonstrated to help networks maintain plasticity in reinforcement [Ball et al., 2023, Lyle et al., 2023] and continual [Kumar et al., 2023] learning. Some additional works have aimed to replicate the benefits of normalization layers via normalization of the network *parameters* [Salimans and Kingma, 2016, Arpit et al., 2016], though layer normalization (LayerNorm) remains standard practice in most domains. A variety of analyses highlight LayerNorm's effect on gradient moments as a critical factor in its efficacy [Xu et al., 2019, Xiong et al., 2020].

## 2.2 Effective learning rates

As noted by several prior works [Van Laarhoven, 2017, Li and Arora, 2020, Li et al., 2020b], normalization of the type performed by BatchNorm and LayerNorm introduces scale-invariance into the layers to which it is applied, where by a scale-invariant function $f$ we mean $f(c\theta, \mathbf{x}) = f(\theta, \mathbf{x})$ for any positive scalar $c > 0$. This leads to the gradient scaling inversely with the parameter norm, i.e. $\nabla f(c\theta) = \frac{1}{c}\nabla f(\theta)$. The intuition behind this property is simple: changing the direction of a large vector requires a greater perturbation than changing the direction of a small vector. This motivates the concept of an 'effective learning rate', which provides a scale-invariant notion of optimizer step size. In the following definition, we take the approach of Kodryan et al. [2022] and assume an implicit 'reference norm' of size 1 for the parameters.

**Definition 1** (Effective learning rate). *Consider a scale-invariant function $f$, parameters $\theta$ and update function $\theta_{t+1} \leftarrow \theta_t + \eta g(\theta_t)$ for some update function $g$. Letting $\rho = \frac{1}{\|\theta\|}$, we then define the effective learning rate $\tilde{\eta}$ as follows:*

$$\tilde{\eta} = \begin{cases} \eta\rho^2, & \text{if } g(\theta_t) = \nabla_\theta f(\theta_t) \\ \eta\rho, & \text{if } g(\theta_t) = \frac{\nabla_\theta f(\theta_t)}{\|\nabla_\theta f(\theta_t)\|} \end{cases} \tag{1}$$

*where, letting $\tilde{\theta} = \theta\frac{1}{\|\theta\|}$ we then have $f(\tilde{\theta} + \tilde{\eta}g(\tilde{\theta})) = f(\theta + \eta g(\theta))$*

Thus by reducing the parameter norm, weight decay can have the dual effect of increasing the effective learning rate [Van Laarhoven, 2017, Hoffer et al., 2018], a property which has been extensively analyzed [Arora et al., 2018, Li and Arora, 2020, Li et al., 2020a]. This perspective justifies the decoupling of weight decay and gradient accumulation in AdamW [Loshchilov and Hutter, 2019], along with the application of learning rate schedules to the weight decay parameter [Xie et al., 2024]. It also motivates scaling the norm of the updates to be proportional to the parameter norm in a variety of optimizers [Liu et al., 2021, You et al., 2017, 2020]. The perspective of updates as rotations of the parameters has been applied by recent analyses of the equilibrium dynamics of optimization in scale-invariant networks trained with weight decay [Wan et al., 2021, Kosson et al., 2024]. More recently, the work of Lobacheva et al. [2021] and Kodryan et al. [2022] has studied the training properties of scale-invariant networks trained with parameters constrained to the unit sphere, a training regime we expand upon in this work. A similar approach, referred to as weight standardization, has been demonstrated to reduce the need for normalization layers in diffusion models [Karras et al., 2024].

## 3 Analysis of normalization layers and plasticity

Although widely used and studied, the precise reasons behind the effectiveness of layer normalization remain mysterious. In this section, we provide some new insights into how normalization can help neural networks to maintain plasticity by facilitating the recovery of saturated nonlinearities, and highlight the importance of controlling the parameter norm in networks which incorporate normalization layers. We leverage these insights to propose Normalize-and-Project, a simple training protocol to maintain important statistics of the layers and gradients throughout training.

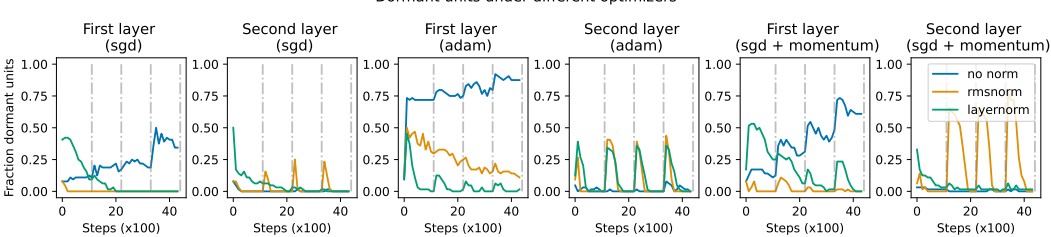

**Figure 1:** Accumulation of dead units in an iterated random label memorization task. The network is trained to memorize random labels of the MNIST dataset which are re-randomized every 1000 optimizer steps. Networks with normalization layers are able to recover from spikes in the number of dead units.

## 3.1 Normalization and ReLU revival

It is widely accepted that achieving approximately mean-zero, unit-variance pre-activations (assuming suitable choices of activation functions) is useful to ensure a network is trainable at initialization [e.g Martens et al., 2021], and many neural network initialization schemes aim to maintain this property as the depth of the network grows. Indeed, it is easy to show that in extreme cases large deviations of these statistics from their initial values can lead to a variety of network pathologies including saturated units and low numerical rank of the empirical neural tangent kernel [Xiao et al., 2020]. Layer normalization not only guarantees that activations are unit-norm, mean-zero at initialization, but also that they stay that way over the course of training even if the data distribution changes, assuming no scale or offset parameters.

Beyond re-normalizing the pre-activation statistics, layer normalization also introduces a dependency between units in a given layer via the mean subtraction and division by standard deviation transformations, which translates to correlations in the gradients of their corresponding weights. This mixing step allows gradients to propagate through a pre-activation even if the unit is saturated, provided layer normalization is applied prior to the nonlinearity, a property we highlight in Proposition 1. We will use the notation $f_{\mathrm{RMS}}(h) = \frac{h}{\|h\|}$ to specify the RMSNorm transform, and let $f'(x) = \frac{\partial}{\partial x}f(x)$ refer to the scalar derivative of any $f$ at scalar $x$.

**Proposition 1.** *Consider two indices $i$ and $j$ of a feature embedding $\phi(f_{\mathrm{RMS}}(h))$ such that $\phi'(f_{\mathrm{RMS}}(h)_j) \neq 0$, and $h_i, h_j \neq 0$. Then we have*

$$\frac{d}{dh_i}\phi(f_{\mathrm{RMS}}(h))_j = -\phi'(f_{\mathrm{RMS}}(h)_j)\frac{1}{\|h\|^3}h_i h_j \neq 0 \, .$$

*In contrast, for post-activation normalization the gradient is zero whenever $\phi'(h_i) = 0$, i.e.*

$$\phi'(h_i) = 0 \implies \frac{d}{dh_i}f_{\mathrm{RMS}}(\phi(h))_j = -\phi'(h_i)\frac{1}{\|\phi(h)\|^3}\phi(h_i)\phi(h_j) = 0. \tag{2}$$

In other words, normalization effectively gives dead ReLU units a second chance at life – rather than immediately decaying to zero, the gradients propagated to the incoming weights of a saturated ReLU will take on non-zero values, which depend on the gradients of the mean and variance of that particular layer. These gradients will be much smaller than those that would typically backpropagate to the unit, but if an optimizer such as Adam or RMSProp is used to correct for the gradient norm, then the unit may still be able to take nontrivial steps, which have a chance at propelling it back into the activated regime. This property is also naturally inherited by layer normalization, which can be viewed as the composition of RMSNorm with a centering transform. An illustration of normalization allowing the network to revive dead units is given in Figure 1. We include the full derivation of Proposition 1 in Appendix A.2, and we demonstrate the effect this can have on a theoretical model of neural network training in Appendix C.4. However, while layer normalization can help the network to recover from saturated nonlinearities, it introduces a new source of potential saturation which must be carefully considered, which is something we will do in the next section.

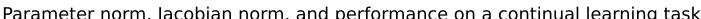

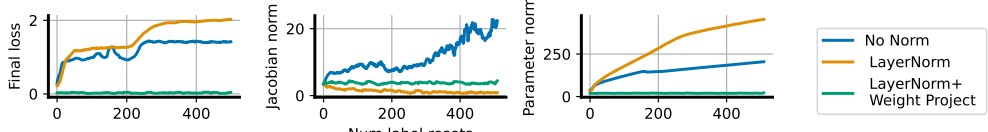

**Figure 2:** Continual random-labels CIFAR training: simple feedforward network architecture (No Norm) exhibits rapid growth in its parameter norm and the norm of its gradients, whereas the otherwise-identical network with layer normalization sees parameter norm growth coupled with a *reduction* in the norm of its gradients and reduced performance on later tasks. Constraining the parameter norm of this network maintains the performance of a random initialization.

## 3.2 Parameter norm and effective learning rate decay

While the *output* of a scale-invariant function is insensitive to scalar multiplication of the parameters, its *gradient* magnitude scales inversely with the parameter norm. Consequently, growth in the norm of the parameters corresponds to a decline in the network's sensitivity to changes in these parameters. In a sense this is preferable, as the glacially slow but stable regime of vanishing gradients is easier to recover from than the unstable exploding gradient regime. However, if the parameter norm grows indefinitely then the corresponding reduction in the effective learning rate will eventually cause noticeable slowdowns in learning – indeed, we show in Figure 2 that it is quite easy to induce this type of situation. To do so, we take a small base convolutional neural network architecture (detailed in Appendix B.4) and train it on random labels of the CIFAR-10 dataset, akin to the classic setting of Zhang et al. [2021a]. We then re-randomize these labels and continue training, repeating this process 500 times. When we apply this process to a network without normalization layers, the Jacobian norm grows to unstable values as the parameter norm increases; in contrast, an equivalent architecture with normalization layers sees a sharp decline in the Jacobian norm as the parameter norm increases. In both cases, the end result is similar: increased parameter norm accompanies reduced performance on new tasks.

While this particular problem is artificial, it is a real and widely observed underlying phenomenon that the magnitudes of neural network parameters tend to increase over the course of training [Nikishin et al., 2022, Abbas et al., 2023]. In a supervised learning problem, where one is using a fixed training budget, the ELR decay induced by growing parameter norms might be desirable and help to protect against too-large learning rates [Arora et al., 2018, Salimans and Kingma, 2016]. Allowed to continue to extremes, however, ELR decay becomes problematic [Lyle et al., 2024]. Fortunately, this problem admits an obvious solution: re-scaling the parameters to have nontrivial ELR. Since LayerNorm induces scale-invariance, this will not change the function computed by the network, but will change its training dynamics. We demonstrate the utility of this strategy in Figure 2.

## 3.3 Normalize-and-Project

We conclude from the above investigation that normalizing a network's pre-activations and fixing the parameter norm presents a simple but effective defense against loss of plasticity. In this section, we propose a principled approach to combine these two steps which we call Normalize-and-Project. Our goal for NaP is to provide a flexible recipe which can be applied to essentially any architecture, and which improves the stability of training, motivated by but not limited to non-stationary problems. Our approach can be decomposed into two steps: the **insertion of normalization layers** prior to nonlinearities in the network architecture, and the **periodic projection** of the network's weights onto a fixed-norm radius throughout training, along with a corresponding update to the per-layer learning rates into the optimization process. Algorithm 1 provides an overview of NaP.

**Layer normalization.** in order to benefit from the robustness to saturated nonlinearities outlined in Proposition 1, we propose adding layer normalization prior to every nonlinearity in the network. While it might seem extreme, this proposal is not too far removed from standard practice. For example, Vaswani et al. [2017] apply normalization after every two fully-connected layers, and recent results suggesting that adding normalization to the key and query matrices in attention heads [Henry et al., 2020] can improve performance and the robustness of optimization [Wortsman et al., 2023].

**Algorithm 1** NaP: Normalize-and-Project

**Input:** network $\mathcal{N}$, input $x$
**for** nonlinearity $\phi_l$ in network **do**
    **if** $\phi_l$ not already normalized **then**
        $\phi_l \leftarrow \phi_l \circ \text{LayerNorm}$
    **end if**
**end for**
compute $\theta' = \texttt{update}(\theta)$
**for** parameter $W_l$ in network **do**
    compute $W_l \leftarrow \text{WeightProject}(W_l)$
**end for**

$\text{WeightProject}(W_l, \rho_l)$ :
**if** $W_l$ is a weight parameter **then**
    $W_l \leftarrow \frac{\rho_l W_l}{\|W_l\|}$
**else**
    $\sigma_l, \mu_l \leftarrow W_l, d \leftarrow \text{len}(\sigma_l)$
    $\sigma_l \leftarrow \sigma_l \sqrt{\frac{d}{\|\sigma_l\|^2 + \|\mu_l\|^2}}$
    $\mu_l \leftarrow \mu_l \sqrt{\frac{d}{\|\sigma_l\|^2 + \|\mu_l\|^2}}$
**end if**

**Step 1:** re-normalize each activation function's inputs in each forward pass

**Step 2:** periodically re-normalize the weights

**Weight projection.** As discussed in Section 3.2, once we have incorporated layer normalization we must take care to ensure that these normalization layers will not saturate, i.e. that the network's effective learning rate will not decay to zero over the course of training due to growth of the parameters. We propose disentangling the parameter norm from the effective learning rate by enforcing a constant norm on the weight parameters of the network, allowing scaling of the layer outputs to depend only on the learnable scale and offset parameters. This approach is similar to that proposed by Kodryan et al. [2022], but importantly takes care to treat the scale and offset parameters separately from weights. For simplicity, we remove bias terms as these are made redundant by the learnable offset parameters. In order to maintain constant parameter norm, we rescale the parameters of each layer to match their initial norm periodically throughout training – the precise frequency is not important as long as the parameter norm does not meaningfully grow to a point of slowing optimization between projections. For example, we find that in Rainbow agents an interval of 1000 steps and 1 step produce nearly identical empirical results. In principle we could constrain the parameter norm to any arbitrary value, but the choice of fixing the initial norm makes learning rate transfer easier when adapting an existing architecture. We do not project the final linear output layer, as it is not scale-invariant.

**Scale and offset parameters.** We find it is absolutely critical to normalize the weight parameters, as these represent the bulk of trainable parameters in the network. The learnable scale and offset parameters, assuming they are included in the network[3], must be dealt with differently. Whereas rescaling the parameters of a linear map that immediately precedes a LayerNorm transform does not change the output of the function, the scale and offset terms will be first passed through a nonlinearity before entering the next LayerNorm and so will not necessarily share this property. For these parameters, we can proceed in one of three ways. In the case of homogeneous activations such as ReLUs, we can normalize the concatenated scale-offset vector as described in in Appendix D. This can require some effort to implement due to the dependency between the scale and offset parameters and may not be worthwhile for small training runs – indeed, most of our empirical results did not require this step, though we include it in our analysis of smaller networks in Figure 3.

A more general solution which requires less implementation overhead and which applies equally to non-ReLU activations is to regularize the scale and offset parameters to their initial values, a strategy which we employ in our continual learning evaluations in Section 5. Assuming suitable initial values, this approach encourages the mean and variance of the pre-activations toward values where the nonlinearity does not saturate. Finally, they can be allowed to drift unconstrained from their initialization, a choice we find unproblematic in standard benchmarks for supervised learning. For a more detailed discussion on this choice, we refer to Appendix D.

---

[3]We observed in many of our experiments that removing trainable scale and offset parameters often has little effect on network performance. In Rainbow agents, for example, removing the trainable offset parameter even improves performance in several environments.

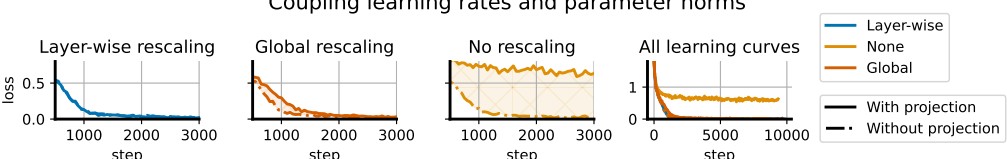

**Figure 3:** We run a 'coupled networks' experiment as described in the text. All networks exhibit similar learning curves, as seen by the rightmost subplot, however there is small but visible gap between the learning curves obtained by NaP and an unconstrained network with fixed learning rates. Using a global learning rate schedule almost entirely closes this gap, but does not induce a precise equivalence in the dynamics as obtained by layer-wise rescaling (leftmost).

## 4 Understanding effective learning rate dynamics

NaP constrains the network's effective learning rate to follow an explicit rather than implicit schedule. In this section, we explore how this property affects network training dynamics, demonstrating how implicit learning rate schedules due to parameter norm growth can be made explicit and be leveraged to improve the performance of NaP in deep RL domains.

### 4.1 Learning rate and parameter norm equivalence

We begin our study of effective learning rates by illustrating how the implicit learning rate schedule induced by the evolution of the parameter norm can be translated to an explicit schedule in NaP. We study a small CNN described in Appendix B.4 with layer normalization prior to each nonlinearity trained on CIFAR-10 with the usual label set. We train two 'twin' scale-invariant networks with the Adam optimizer in tandem: both networks see the exact same data stream and start from the same initialization, but the per-layer weights of one are projected after every gradient step to have constant norm, while the other is allowed to vary the norms of the weights. We then consider three experimental settings: in the first, we re-scale the per-layer learning rates of the projected network so that the explicit learning rate is equal to the effective learning rate of its twin. In the second, we re-scale the global learning rate based on the ratio of parameter norms between the projected and unprojected network, but do not tune per-layer. In the third, we do no learning rate re-scaling. We see in Figure 3 that the shapes of the learning curves for all networks except for the constant-ELR variant are quite similar, with the global learning rate scaling strategy producing a smaller gap than the no-rescaling strategy. By construction, the dynamics of the per-layer rescaling network and its twin are identical. Because global learning rate schedules are standard practice and induce dynamics that are quite close to those obtained by parameter norm growth in Figure 3, we take this approach in the remainder of the paper, leaving layer-specific learning rates and schedules for future work.

### 4.2 Implicit learning rate schedules in deep RL

When taken to extremes, learning rate decay will eventually prevent the network from making nontrivial learning progress. However, learning rate decay plays an integral role in the training of many modern architectures, and is required to achieve convergence for stochastic training objectives (unless the interpolation applies or Polyak averaging is employed). In this section we will show that, perhaps unsurprisingly, naive application of NaP with a constant effective learning rate can sometimes harm performance in settings where the implicit learning rate schedule induced by parameter norm growth was in fact critical to the optimization process. More surprising is that the domain where this phenomenon is most apparent is one where common wisdom would suggest learning rate decay would be undesirable: value-based deep reinforcement learning.

RL involves a high degree of nonstationarity. As a result algorithms such as DQN and Rainbow often use a constant learning rate schedule. Given that layer normalization has been widely observed to improve performance in these algorithms, and that parameter norm tends to increase significantly in deep RL tasks, one might at first believe that the performance improvement offered by layer normalization is happening in *spite* of the resulting implicit learning rate decay. A wider view of the literature, however, reveals that several well-known algorithms such as AlphaZero [Schrittwieser et al., 2020], along with many implementations of popular methods such as Proximal Policy Opti-

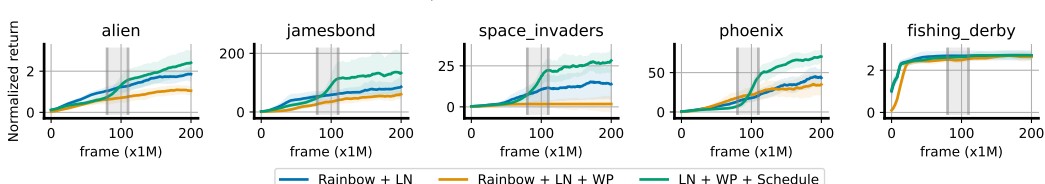

**Figure 4:** Without an explicit learning rate schedule, a Rainbow trained with NaP may fail to make any performance improvement; while the implicit schedule induced by the parameter norm is clearly important to performance, in several games this is significantly outperformed by a simple linear schedule terminating halfway through training. Intriguingly, we see a characteristic sharp improvement near the end of the decay schedule in several (though not all, e.g. fishing derby) games.

mization [Schulman et al., 2017], incorporate some form of learning rate decay, suggesting that a constant learning rate is not always desirable. Indeed, Figure 4 shows that constraining the parameter norm to induce a fixed ELR in the Rainbow agent frequently results in *worse* performance compared to unconstrained parameters. This is particularly striking given that many of the benefits supposedly provided by layer normalization, such as better conditioning of the loss landscape [Lyle et al., 2023] and mitigation of overestimation bias [Ball et al., 2023], should be independent of the effective learning rate. Instead, these properties appear to either be irrelevant for optimization or dependent on reductions in the effective learning rate to materialize.

We can close this gap by introducing a learning rate schedule (linear decay from the default $6.25 \cdot 10^{-5}$ to $10^{-6}$, roughly proportional to the average parameter norm growth across games). We further observe in Appendix C.1 that when we vary the endpoint of the learning rate schedule, we often obtain a corresponding x-axis shift in the learning curves, suggesting that reaching a particular learning rate was necessary to master some aspect of the game. We conclude that, while beneficial, the implicit schedule induced by the parameter norm is not necessarily *optimal* for deep RL agents, and it is possible that a more principled adaptive approach could provide still further improvements.

## 5    Empirical Evaluations

We now validate the utility of NaP empirically. Our goal in this section is to validate two key properties: first, that NaP does not hurt performance on stationary tasks; second, that NaP can mitigate plasticity loss under a variety of both synthetic and natural nonstationarities.

### 5.1    Robustness to nonstationarity

We begin with a continual classification problem which has been widely used in works studying loss of plasticity: memorization of iteratively re-randomized labels of an image dataset. In this task, each input from the dataset is assigned a uniform random label; the network is then trained on this set of labels for a fixed duration, after which a new set of random labels are generated and optimization begins again. Full details can be found in Appendix B.4. There is no shared structure between tasks in this problem setting, so performance is solely determined by trainability and not transfer between tasks. We evaluate our approach on a variety of sources of nonstationarity, using two architectures: a small CNN, and a fully-connected MLP (see Appendix B.4. for details). We consider a number of methods designed to maintain plasticity including Regenerative regularization [Kumar et al., 2023], Shrink and Perturb [Ash and Adams, 2020], ReDo [Sokar et al., 2023], leaky ReLU units (inspired by the success of concatenated ReLU activations [Abbas et al., 2023]), L2 regularization, and random Gaussian perturbations to the optimizer update, a heuristic form of Langevin Dynamics. We track the average online accuracy over the course of training for 20M steps, equivalent to 200 data relabelings, using a constant learning rate. We find varying degrees of efficacy in these approaches, with regenerative regularization and ReDO tending to perform the best. When we apply NaP on top of the same suite of methods in Figure 5, we observe near-monotonic improvements (with the exception of ReDO, where the reset mechanism does not take normalization into account) in performance and a significant reduction in the gaps between methods, with the performance curves of the different methods nearly indistinguishable in the MLP. Further, we observe constant or increasing slopes in

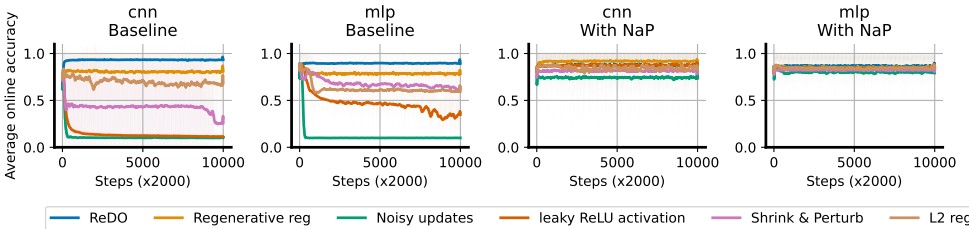

**Figure 5: Robustness to nonstationarity:** we see that without NaP, there is a wide spread in the effectiveness of various plasticity-preserving methods across two architectures. Once we incorporate NaP, however, the gaps between these methods shrink significantly and almost uniformly improves over the unconstrained baseline.

|  | CIFAR-10 | ImageNet-1k | C4 | Pile | WikiText | Lambada | SIQA | PIQA |
|---|---|---|---|---|---|---|---|---|
| NaP | **94.64** | 77.26 | **45.7** | **47.9** | **45.4** | **56.6** | **44.2** | **68.8** |
| Baseline | **94.65** | 77.08 | 44.8 | 47.4 | 44.2 | 54.1 | 43.5 | 67.3 |
| Norm only | 94.47 | **77.45** | 44.9 | 47.6 | 44.3 | 53.6 | 43.8 | 67.1 |

**Table 1: Left:** Top-1 prediction accuracy on the test sets of CIFAR-10 and ImageNet-1k. **Right:** per-token accuracy of a 400M transformer model pretrained on the C4 dataset, evaluated on a variety of language benchmarks. See Appendix C.5 for more results with variation measures.

the online accuracy, suggesting that the difference between methods has more to do with their effect on within-task performance than on plasticity loss once the parameter and layer norms have been constrained.

## 5.2 Stationary supervised benchmarks

Having observed remarkable improvements in synthetic tasks, we now confirm that NaP does not interfere with learning on more widely-studied, natural datasets.

**Large-scale image classification.** We begin by studying the effect of NaP on two well-established benchmarks: a VGG16-like network [Simonyan and Zisserman, 2014] on CIFAR-10, and a ResNet-50 [He et al., 2016] on the ImageNet-1k dataset. We provide full details in Appendix B.4. In Table 1 we obtain comparable performance in both cases using the same learning rate schedule as the baseline.

**Natural language:** we evaluate the effect of NaP on a 400M-parameter transformer architecture (details in Appendix B.3) trained on the C4 dataset [Raffel et al., 2020]. Table 1 shows that our approach does not interfere with performance on this task, where we match final per-token accuracy of the baseline. When evaluating the pre-trained network on a variety of other datasets, we find that NaP slightly outperforms baselines in terms of performance on a variety of benchmarks, including WikiText-103, Lambada [Paperno et al., 2016], PIQA [Bisk et al., 2020], SocialIQA [Sap et al., 2019], and Pile [Gao et al., 2020].

## 5.3 Deep reinforcement learning

Finally, we evaluate our approach on a setting where maintaining plasticity is critical to performance: RL on the Arcade Learning Environment. We conduct a full sweep over 57 Atari 2600 games comparing the effects of normalization, weight projection, and learning rate schedules on a Rainbow agent [Hessel et al., 2018]. In the RHS of Figure 6 we plot the spread of scores, along with estimates of the Mean and IQM of four agents: standard Rainbow, Rainbow + LayerNorm, Rainbow + NaP without an explicit LR schedule, and Rainbow + NaP with the LR schedule described in Section 4.2. We find that NaP with a linear schedule outperforms the other methods.

We also consider the sequential setting of Abbas et al. [2023]. In this case, we consider an idealized setup where we reset the optimizer state and schedule every time the environment changes, using a cosine schedule with warmup described in Appendix B.2. To evaluate NaP on this regime, we train on each of 10 games for 20M frames, going through this cycle twice. We do not reset parameters of the continual agents between games, but do reset the optimizer. We plot learning curves for the

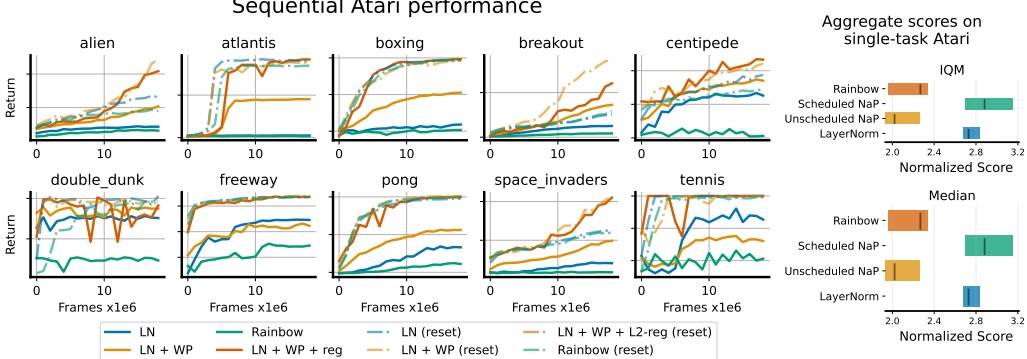

**Figure 6: Left:** We visualize the learning curves of continual atari agents on sequential ALE training (i.e. 200M frames). Each game is played for 20M frames, and agents pass sequentially from one to another, repeating all ten games twice for a total of 400M training frames. Solid lines indicate performance on the second visit to each game, and dotted lines indicate performance of a randomly initialized network on the game. Even in its second visit to each game, NaP performs comparably the randomly initialized networks, whereas the standard rainbow agent exhibits poor performance on all games in the sequential training regime. **Right:** aggregate effects of normalization on single-task atari, computed via the approach of Agarwal et al. [2021]. Bars indicate 95% confidence intervals over 4 seeds and 57 environments.

second round of games in the LHS of Figure 6, finding that NaP significantly outperforms a baseline Rainbow agent with and without layer normalization. Indeed, even after 200M steps the networks trained with NaP make similar learning progress to a random initialization.

## 6   Discussion

This paper has demonstrated that maintaining plasticity in the face of nonstationary training objectives can be achieved through careful normalization of the network's features and parameters. While there are many factors contributing to the efficacy of normalization in maintaining plasticity, we identified two non-obvious factors: the effect of normalization on the network's ability to revive saturated units, and role of the parameter norm in determining the effective learning rate of networks with normalization layers. With these insights in hand, we proposed Normalize-and-Project, a simple protocol which consists of adding layer normalization prior to nonlinearities in the network and periodically re-scaling the per-layer weights back to their initial norms.

Beyond improving performance in non-stationary supervised learning problems, NaP also presents a powerful tool for understanding the role of the effective learning rate on training dynamics. By applying NaP to reinforcement learning problems, we revealed the crucial importance of ELR decay to the ability of value-based deep RL agents to improve their performance on Atari tasks, explaining why approaches such as weight decay so often struggle to provide the same performance benefits in reinforcement learning as they do in supervised problems. This finding opens the door to a number of further questions: *why* is ELR decay so critical to performance RL? What features of the environments require a sufficiently low ELR to learn, and why? In what settings is the implicit learning rate schedule yielded by the parameter norm sub-optimal for learning progress, and can better schedules be determined automatically, rather than being proscribed prior to the start of training?

The development of even more effective normalization strategies is a further promising avenue for future work. While we did not observe pathological behaviour in the unnormalized and unregularized scale and offset parameters of networks trained on single tasks, regularization of these parameters was critical to maintain performance in the sequential Atari domain and suggests that these parameters can interfere with learning if left unconstrained. Further, while layer normalization is not observed to impede network expressivity in visual domains, where the scale of the input does not usually carry task-relevant information, in proprioceptive domains normalization of features may increase the difficulty of learning, as it removes valuable spatial information from the input. Applying NaP to these domains will require careful design of feature embedding layers or an alternate normalization strategy which does not remove information about the Euclidean distance between inputs.

## Acknowledgements

We would like to thank Vincent Roulet for his helpful and detailed feedback on earlier versions of this manuscript.

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

# A Derivations

## A.1 Notation

Analysis of a network's training dynamics depends on characterizing the evolution of (pre-)activations and gradients in the forward and backward passes respectively. We lay out basic notation for fully-connected layers first, and then note additional details which must be considered for convolutional, skip connection, and attention layers. We will write the parameters of a layer as $\theta_l$, and use $f$ to denote a neural network.

**Fully-connected layers:** We write the forward pass through a network $f : \mathbb{R}^{d_0} \to \mathbb{R}^{d_L}$ as a composition of layer-wise computations $f^l : \mathbb{R}^{d_{l-1}} \to \mathbb{R}^{d_l}$ of the form:

$$a^l = f^l(a^{l-1}) = \phi(\sigma^l f_{\mathrm{LN}}(h^l) + \mu^l), \quad h^l = W^l a^{l-1} \; W^l = \theta_l \tag{3}$$

where $\phi$ denotes a nonlinearity and $f_{\mathrm{LN}}$ is a (possibly absent) normalization operator. We let $a^0$ denote the network inputs. We will refer to a forward pass through a subset of a network with the notation $f^{l_1:l_2} = f^{l_2} \circ \cdots \circ f^{l_1}$, and use interchangeably $f = f^{1:L}$. We will refer to the full set of network parameters by $\theta = \mathrm{vec}(\{\theta^l\}_{l=1}^L)$.

**Convolutional layers:** since a convolution can be viewed as a parameterization of a matrix with a particular symmetry, we express these layers identically to the fully-connected layers, with a change in semantics such that $W^l$ is the matrix representation of the convolutional parameters $\theta_l$, where we write the embedding of $\theta_l$ into a matrix as $W_{\mathrm{conv}}(\theta_l)$. For simplicity, we ignore the choice of padding.

$$\phi(\sigma^l f_{\mathrm{LN}}(h^l) + \mu^l), \quad h^l = W^l a^{l-1} \text{ and } W^l = W_{\mathrm{conv}}(\theta_l) \tag{4}$$

**Skip-connect layers:** in some network architectures, a nonlinearity is placed on the outputs of two subnetworks to produce a function of the form

$$f^l = \phi\left(\sum_{l_i \in L} a_{l_i}\right) \quad \text{or} \quad f^l = \phi\left(f_{\mathrm{LN}}\left(\sum_{l_i \in L} a_{l_i}\right)\right) \tag{5}$$

for some index set $L$. The choice of whether to apply normalization to the sum of the subnetwork outputs or to each output individually depends on the desired signal propagation properties of the network [De and Smith, 2020].

## A.2 Derivation of Proposition 1

The result described in proposition 1 follows straightforwardly from the chain rule. For pre-activation RMSNorm we have

$$\frac{d}{dh_i}\phi(f_{\mathrm{RMS}}(h))_j = \phi'(f_{\mathrm{RMS}}(h))_j \frac{d}{dh_i} f_{\mathrm{RMS}}(h)_j \tag{6}$$

$$\frac{d}{dh_i} f_{\mathrm{RMS}}(h)_j = \frac{d}{dh_i} \frac{h_j}{\left(\sum h_k^2\right)^{1/2}} \tag{7}$$

$$= -\frac{1}{2} \frac{h_j}{\left(\sum h_k^2\right)^{3/2}} \frac{d}{dh_i} \sum h_k^2 \tag{8}$$

$$= -\frac{h_j}{\left(\sum h_k^2\right)^{3/2}} h_i \tag{9}$$

$$\frac{d}{dh_i}\phi(f_{\mathrm{RMS}}(h))_j = -\frac{\phi'(f_{\mathrm{RMS}}(h))_j h_j h_i}{\|h\|^3} \tag{10}$$

## A.3 Gradients and signal propagation

One perspective which can provide some additional insight into our approach is to consider the following decomposition of the gradient being backpropagated through a layer.

$$\nabla_{W_k}\mathcal{L}(\theta) = \partial_{a_k}\mathcal{L}(\theta) \cdot \partial_{W_k}\phi(f_{\mathrm{LN}}(W_k a_{k-1})) \tag{11}$$

$$= \partial_{a_k}\mathcal{L}(\theta) \cdot D_{\dot{\phi}} \nabla_{h_k} f_{\mathrm{LN}}(h_k)) a_{k-1}^{\top} \tag{12}$$

With this decomposition, we obtain the following interpretation of some common pathologies.

**Saturated nonlinearities:** a saturated nonlinearity implies that $D_{\dot{\phi}}$, and the gradient is exactly (in the case of ReLU) or very close to (e.g. tanh) zero. As a result, $u_t$ will be zero for the affected coordinates and the corresponding parameters will remain frozen. NaP addresses this problem by using Layer or RMSNorm prior to any nonlinearity in the network.

**Saturated normalization layers:** the normalization transformation $x \mapsto \frac{x}{\|x\|}$ is vulnerable to saturation as $\|x\|$ grows, in the sense that for a fixed-norm update $u$ we will have

$$\lim_{\|x\|\to\infty} \frac{x+u}{\|x+u\|} - \frac{x}{\|x\|} = 0 \tag{13}$$

and so the term $\nabla_{h_k} f_{\mathrm{LN}}(h_k))$ will vanish. In this case, while the optimizer will be able to update the parameters, these updates will have a diminishing effect on the network's output.

**Vanishing and exploding gradients:** divergence and disappearance of the activations $a_{k-1}$ and backpropagated gradients $\partial_{a_k}\mathcal{L}(\theta)$ are well-known pathologies which can make networks untrainable. However, even networks which start training from a well-tuned initialization may still encounter exploding gradients due to parameter norm growth over time [Dohare et al., 2021, Wortsman et al., 2023], or vanishing gradients and activations, such as in the case of saturated (i.e. dead/dormant) ReLU units [Sokar et al., 2023].

## A.4 Details of NaP

**Guiding principles:** in general, the goal of NaP is to avoid dramatic distribution shifts in the pre-activation and parameter norms, and to ensure that the network can perform updates to parameters even if a nonlinearity is saturated. With these in mind, there are two key properties that a network designer should aim to maintain:

1. All **parametric** functions entering a nonlinearity should have a normalization layer that at ensures the gradients of all units' parameters are correlated. If there are no parameters between nonlinearities (as is sometimes the case in e.g. resnets) normalization is not essential.

2. Based on our investigations in Appendix C.4, *L2 normalization* of the pre-activations is crucial to obtain the positive benefits of layer normalization, while centering does not have noticeable effects on the network's robustness to unit saturation. As a result, applying at least RMSNorm is crucial prior to nonlinearities, but the choice of whether or not to incorporate centering is up to the designer's discretion.

**Batch normalization layers:** by default, we put layer normalization prior to batch normalization if an architecture already incorporates batch normalization prior to a nonlinearity. This preserves the property of batch norm that individual units have mean zero across the batch, which may not be the case if layer normalization is applied after. We also always omit offset parameters if layernorm is succeeded by batchnorm, as these offset parameters will be zeroed out by batchnorm.

**Skip-connect layers:** provided that layer normalization is applied to the outputs of a linear transformation prior to a nonlinearity, NaP is agnostic to whether normalization is applied prior to or after a residual connection's outputs are added to the output of a layer. In particular, if we have a layer of the form $\phi(a_1 + a_2)$ and $a_1$ and $a_2$ are the outputs of some subnetwork of the form $\phi_1(f_{\mathrm{LN}}(h_1))$ and $\phi_2(f_{\mathrm{LN}}(h_2))$ where $\phi_1$ and $\phi_2$ are (possibly trivial) activation functions, then the relevant parameters will already benefit from Proposition 1 and it is not necessary to add an additional normalization layer prior to the activation $\phi$.

**Attention layers:** unlike linear layers, attention layers do not typically include bias terms. To analogize this common practice in NaP, we omit the offset and mean subtraction components of the LayerNorm transform, obtaining

$$\mathrm{MHA}(W_Q X, W_K X) \mapsto \mathrm{MHA}(\sigma_Q f_{\mathrm{RMS}}(W_Q X), \sigma_K f_{\mathrm{RMS}}(W_K X)) \tag{14}$$

where crucially $f_{\mathrm{RMS}}$ is not applied along the token axis as this can lead to leakage of information during training on next-token-prediction objectives. A similar problem also prevents us from using normalization directly on the QK product matrix, along with the observation that the intuition of normalizing vectors so that their dot product is equal to the cosine similarity is lost once the dot products have already occurred [Henry et al., 2020]. Empirically, we find that the scale parameters $\sigma_Q$ and $\sigma_K$ don't seem to be strictly necessary for expressivity, and that networks can even form selective attention masks for in-context learning without using these parameters to further saturate the softmax.

## A.5 Dynamics of NaP

**Weight projection non-interference:** NaP incorporates a projection onto the ball of constant norm after each update step. A natural question is whether this projection step might simply be the inverse of the update step, leaving the parameters of the network constant. Fortunately, we note that the normalization layers have the effect of projecting gradients onto a subspace which is orthogonal to the current parameter values, i.e.

$$\left( \nabla_{\mathbf{x}} f_{\mathrm{RMS}}(\mathbf{x}) \right)(\mathbf{x}) = \mathbf{0} \, . \tag{15}$$

We also note that except for extreme situations such as Neural Collapse [Papyan et al., 2020], real-world gradient updates are almost never colinear with the parameters, meaning that even without normalization layers this problem would be unlikely.

Another concern that arises from the constraints we place on the weights and features is the possibility that these constraints will limit the network's expressivity. Normalization does remove the ability to distinguish colinear inputs of differing norms, meaning that the inputs $\mathbf{x}$ and $\alpha \mathbf{x}$ will map to the same output for all $\alpha$; however, since many data preprocessing pipelines already normalize inputs, we argue this is not a significant limitation. Indeed, under a more widely-used notion of expressivity, the number of *activation patterns* [Raghu et al., 2017], NaP does not limit expressivity at all. While straightforward, we provide a formal statement and proof of this claim in Appendix A.7.

**Layer normalization and parameter growth:** In fact, if we incorporate normalization layers into the network we might expect an even more aggressive decay schedule. Recall that in a scale invariant network, we have $\langle \nabla_\theta f(\theta), \theta \rangle = 0$. Thus we know that the gradient at each time step will be orthogonal to the current parameters. In an idealized setting where we use the update rule $\theta_{t+1} \leftarrow \theta_t + \alpha \frac{\nabla_\theta \ell(\theta_t)}{\|\nabla_\theta \ell(\theta_t)\|}$, this would result in the parameter norm growing at a rate $\Theta(t)$, corresponding to an effectively linear learning rate decay.

## A.6 Scale-invariance and layer-wise gradient norms

One benefit of NaP is that, because we normalize layer outputs, we limit the extent to which divergence in the norm of one layer's parameters can propagate to the gradients of other layers. For e.g. linear homogeneous activations such as ReLUs, the gradient of some objective function for some input with respect to the parameters of a particular layer contains a sum of matrix products whose norm will depend multilinearly on the norm of each matrix. In particular, in the simplified setting of a deep linear network where $f(\theta, \mathbf{x}) = \prod W^l \mathbf{x}$, we recall Saxe et al. [2013]

$$\nabla_{W^l} f(\theta; \mathbf{x}) = \left[ \prod_{k>l} W^k \right]^\top \mathbf{x}^\top \left[ \prod_{k<l} W^k \right]^\top \tag{16}$$

In particular, with $\theta' = W^1, \ldots, cW^k, \ldots, W^L$, for $k \neq l$ we would have

$$\implies \nabla_{W^l} f(\theta'; \mathbf{x}) = c \nabla_{W^l} f(\theta; \mathbf{x}) \tag{17}$$

The situation changes little if we add ReLU nonlinearities to the network. In this case, we use the notation $\mathbf{D}_{\phi_l}(\mathbf{x})$ to denote the diagonal matrix indicating whether $a^l[i] > 0$

$$\nabla_{W^l} f(\theta; \mathbf{x}) = \left[ \prod_{k>l} \mathbf{D}_{\phi_k}(\mathbf{x}) W^k \right]^\top \mathbf{x}^\top \left[ \prod_{k<l} \mathbf{D}_{\phi_k}(\mathbf{x}) W^k \right]^\top \tag{18}$$

$$\implies \nabla_{W^l} f(\theta'; \mathbf{x}) = c \nabla_{W^l} f(\theta; \mathbf{x}) \tag{19}$$

If we incorporate a normalization layer at the end of the network (for simplicity we consider RMSNorm here, but a similar argument applies to standard LayerNorm), the scale-invariance of the resulting output means that the norms of each layer's gradients are independent of the norms of the other layers' parameters, i.e.

$$\nabla_{W^l} f_{\mathrm{RMS}} \circ f(\theta; \mathbf{x}) = \nabla_{W^l} f_{\mathrm{RMS}} \circ f(\theta'; \mathbf{x}) \ \ \text{whenever} \ \theta' = (W_1, \ldots, cW_k, \ldots, W_L), k \neq l \tag{20}$$

This property is appealing as it means that growth or decay of the norm of a single layer will not interfere with the dynamics of the others. However, it does mean that a layer's effective learning rate will still be sensitive to scaling, which motivates our use of renormalization. It also does not help to avoid saturated units, motivating our use of layer normalization prior to nonlinearities.

### A.7 Expressivity of NaP

Finally, we discuss the effect of normalization and weight projection on a notion of expressivity known as the number of activation patterns [Raghu et al., 2017] exhibited by a neural network. This quantity relates to the complexity of the function class a network can compute, giving the following result the corollary that NaP doesn't interfere with this notion of expressivity.

**Proposition 2.** *Let $f$ be a fully-connected network with ReLU nonlinearities. Let $\tilde{f}$ be the function computed by $f$ after applying NaP. Then the activation pattern of a particular architecture $f$ and parameter $\theta$ be $\mathcal{A}_\theta$, we have*

$$\mathcal{A}_f(\theta, \mathbf{x}) = \mathcal{A}_{\tilde{f}}(N(\theta), \mathbf{x}) . \tag{21}$$

*Further, the decision boundary $\max_{i \in d_{out}} f_i(\mathbf{x})$ is preserved under NaP.*

*Proof.* We apply an inductive argument on each layer. In particular, when $\phi$ is a ReLU nonlinearity we have

$$\mathcal{A}(\phi(\mathbf{h})) = \mathcal{A}(\phi(f_{\mathrm{RMS}}(\mathbf{h})))$$

$$\phi(f_{\mathrm{RMS}}(\mathbf{h})) = \frac{\phi(\mathbf{h})}{\|\mathbf{h}\|}$$

$$\tilde{f}(\mathbf{x}) = \frac{1}{\Pi_{l=1}^L \|\mathbf{h}_l(\mathbf{x})\|} f(\mathbf{x})$$

which trivially results in identical activation patterns in the normalized and unnormalized networks. It is worth noting that one distinguishing factor from a standard ReLU network is that the resulting scaling factor will be different for each $\mathbf{x}$. Thus while the activation patterns will be the same, the two different inputs $\mathbf{x}$ $\mathbf{y}$ might have different scaling factors, which will be a nonlinear function of the input. NaP networks, even with ReLU activations, thus do not have the property of being piecewise linear. $\qquad\square$

### A.8 Rescaling scale/offset parameters (linear homogeneous networks)

We observe that for any $c > 0$, letting $\phi(x) = \max(x, 0)$ we have:

$$f_{\mathrm{LN}}(\phi(W\sigma\mathbf{x} + \mu)) = f_{\mathrm{LN}}(c\phi(W\sigma\mathbf{x} + \mu)) \tag{22}$$

$$= f_{\mathrm{LN}}(\phi(cW(\sigma\mathbf{x} + \mu))) \ \text{by homogeneity of ReLU} \tag{23}$$

$$= f_{\mathrm{LN}}(\phi(W(c\sigma\mathbf{x} + c\mu))) \tag{24}$$

Then we obtain analogous effective learning rates, letting

$$g_{c\mu} = \nabla_\mu f(c\sigma, c\mu; \mathbf{x}) \quad g_{c\sigma} = \nabla_\sigma f(c\sigma, c\mu; \mathbf{x}) \tag{25}$$

we then have equivalent updates for $\|\sigma^2 + \mu^2\| = 1$ and $\eta_c = c^2$

$$f_{\mathrm{LN}}((c\sigma + \eta_c g_{c\sigma})\mathbf{x} + c\mu + \eta_c g_{c\mu}) = f_{\mathrm{LN}}((c\sigma + c^2 g_{c\sigma})\mathbf{x} + c\mu + c^2 g_{c\mu}) \tag{26}$$

$$= f_{\mathrm{LN}}((c\sigma + c^2 \frac{1}{c} g_\sigma + c\mu + c^2 \frac{1}{c} g_\mu) \tag{27}$$

$$= f_{\mathrm{LN}}(c((\sigma + g_\sigma)\mathbf{x} + \mu + g_\mu) = f_{\mathrm{LN}}((\sigma + g_\sigma)\mathbf{x} + \mu + g_\mu) \tag{28}$$

Finally, we note that the above also holds if we apply a linear transformation $W$ to the output of the scale-offset transform, since

$$f_{\mathrm{LN}}(W(c\sigma\mathbf{x} + c\mu)) = f_{\mathrm{LN}}(cW(\sigma\mathbf{x} + \mu)) = f_{\mathrm{LN}}(W(\sigma\mathbf{x} + \mu)) \tag{29}$$

and so the effective learning rate scales precisely as we had previously for linear layers, but now with respect to the joint norm of the scale and offset parameters $\mu, \sigma$.

# B  Experiment details

## B.1  Toy experiment details

We conduct a variety of illustrative experiments on toy problem settings and small networks.

**Network:** in Figures 1 and 2, we use a DQN-style network which consists of two sets of two convolutional layers with 32 and 64 channels respectively. We then apply max pooling and flatten the output, feeding through a 512-unit hidden linear layer before applying a final linear transform to obtain the output logits. The network uses ReLU nonlinearities. When NaP is applied, we add layer normalization prior to each nonlinearity.

## B.2  RL details

**Single-task atari:** We base our RL experiments off of the publicly available implementation of the Rainbow agent [Hessel et al., 2018] in DQN Zoo [Quan and Ostrovski, 2020]. We follow the default hyperparameters detailed in this codebase. In our implementation, we add normalization layers prior to each nonlinearity except for the final softmax. We train for 200M frames on the Atari 57 suite [Bellemare et al., 2013]. We also allow for a learning rate schedule, which we explicitly detail in cases where non-constant learning rates are used.

**Sequential ALE:** we use the same rainbow implementation as for the single-task results, using a cosine decay learning rate for all variants. We restart the cosine decay schedule at every task change for all agents. Our cosine decay schedule uses an init value of $10^{-8}$, a peak value of the default LR for Rainbow (0.000625), 1000 warmup steps after the optimizer is reset, and end-value equal to $10^{-6}$ as in the single-task settings. We choose cosine decay due to its popularity in supervised learning, and to highlight the versatility of NaP to different LR schedules. We follow the game sequence used by Abbas et al. [2023], training for 20M frames per game.

## B.3  Language details

**Sequence memorization:** we set a dataset size of 1024 and a sequence length of 512. We use a vocabulary size of 256, equivalent to ASCII tokenization. We use the adam optimizer, and train all networks for a minimum of 10 000 steps. We reset the dataset every 1000 optimizer steps, generating a new set of 1024 random strings of length 512. We use as a baseline a transformer architecture [Vaswani et al., 2017, Raffel et al., 2020] consisting of 4 attention blocks, with 8 heads and $d_{\mathrm{model}}$ equal to 256. We use a batch size of 128.

**In-context learning:** our in-context learning experiments use the same overall setup as the sequence memorization experiments, with identical architectures and baseline optimization algorithm. In this

case we train on a dataset consisting of 4096 randomly generated strings, in which the final 100 tokens are a contiguous subsequence of the first 412, selected uniformly at random from indices in [1, 312].

**Natural language:** we run our natural language experiments on a 400M parameter transformer architecture based on the same backbone as the previous two tasks, this time consisting of 12 blocks with 12 heads and model dimension 1536. We use the standard practice of learning rate warmup followed by cosine decay, setting a peak learning rate of $2 \cdot 10^{-4}$ which is reached after a linear warmup of 1000 steps. We use a batch size of 128 and train for 30,000 steps. We use a weight decay parameter of 0.1 with the adamw optimizer.

## B.4   Vision details

**CIFAR-10 Memorization:** We consider three classes of network architecture: a fully-connected multilayer perceptron (MLP), for which we default to a width of 512 and depth of 4 in our evaluations; a convolutional network with $k$ convolutional layers followed by two fully connected layers, for which we default to depth four, 32 channels, and fully-connected hidden layer width of 256. In all networks, we apply layer normalization before batch normalization if both are used at the same time. By default, we typically do not use batch normalization. We use ReLU activations

Our continual supervised learning domain is constructed from the CIFAR-10 dataset, from which we construct a family of continual classification problems. Each continual classification problem is characterized by a transformation function on the labels. For label transformations, we permute classes (for example, all images with the label 5 will be re-assigned the label 2), and random label assignment, where each input is uniformly at random assigned a new label independent of its class in the underlying classification dataset. Figures in the main paper concern random label assignments, as this is a more challenging task which produces more pronounced effects.

In our figures in the main paper, we run a total of 20M steps and a total of 200 random target resets. All networks are trained using the Adam optimizer. We conducted a sweep over learning rates for the different architectures, settling on $10^{-4}$ as this provided a reasonable balance between convergence speed and stability in all architectures, to ensure that all networks could at least solve the single-task version of each label and target transformation.

**VGGNet and ResNet-50 baselines:** we use the standard data augmentation policies for the CIFAR-10 and ImageNet-1k experiments. In our ImageNet-1k experiments, we use the ResNetv2 architecture variant, with a label smoothing parameter of 0.1, weight decay $10^{-4}$, and as an optimizer we use SGD with a cosine annealing learning rate schedule, and Nesterov momentum with decay rate 0.9. We use a batch size of 256. Our CIFAR-10 experiments use a VGG-Net architecture. We use the sgd optimizer with a batch size of 32, and set a learning rate schedule which starts at 0.025 and decays by a factor of 0.1 iteratively through training. We use Nesterov momentum with decay rate 0.9. We train for a total of 400 000 steps.

## C   Additional experimental results

### C.1   Arcade learning environment

Our choice of learning rate schedule in the paper is motivated by an attempt to roughly approximate the shape of the implicit schedule obtained by parameter growth on average across games in the suite, with a slightly smaller terminal point than would typically be achieved by the parameter norm alone. We consider learning rate schedules which linearly interpolate between the initial learning rate of 0.000625 and a learning rate of $10^{-6}$, which is roughly equivalent to increasing the parameter norm by a factor of 60. We explore the importance of the duration of this decay in Figure 7, where we conduct a sweep over a subset of the full Atari 57 suite (selected by sorting alphabetically and selecting every third environment). We observe that faster decay schedules result in better performance initially, but often plateau at a lower value. Decaying linearly over the entire course of training, in contrast, exhibits slow initial progress but often picks up significantly towards the end of training. We conclude that in many games, reducing the learning rate is necessary for performance improvements in this agent, and the linear decay over the entire training period doesn't give the agent sufficien time to take advantage of the finer-grained updates to its predictions that a lower learning rate affords.

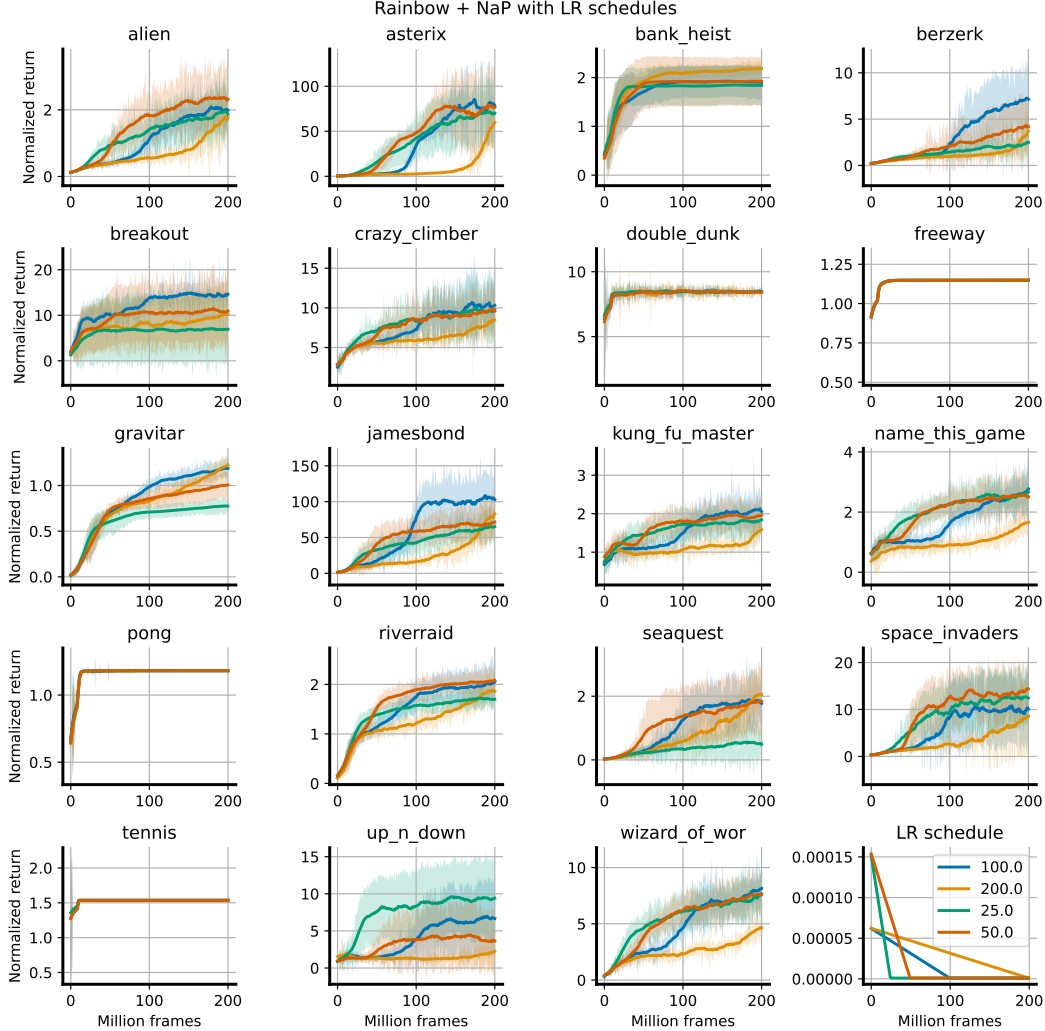

**Figure 7:** We see that faster LR decays typically accompany fast initial progress followed by plateaus. Terminating the linear schedule halfway through training strikes the best balance of the four settings considered for overall progress.

## C.2 Non-stationary MNIST

We include additional network statistics from the experiments shown in Figure 5 in Figure 8.

**Linearized units:** given a large batch, what fraction of the ReLU units in the penultimate layer are either 0 for all units or nonzero for all units. This gives a slightly more nuanced take on the amount of computation performed in the penultimate layer than the feature rank.

**Feature rank:** we compute the numerical rank of the penultimate layer features by sampling a batch of data and computing the singular values of the $b \times d$ matrix of $d-$dimensional feature vectors. $\sigma_1, \ldots, \sigma_d$. We then compute the numerical rank as $\sum \mathbb{1}(\sigma_i/\sigma_1 > 0.01)$.

**Parameter and gradient norm:** these are both computed in the standard way by flattening out the set of parameters / per-parameter gradients and computing the 2-norm of this vector.

## C.3 Non-stationary sequence modeling

We find additionally that NaP is capable of improving the robustness of sequence models to nonstationarities, while also not interfering with the formation of in-context learning circuits. We demonstrate

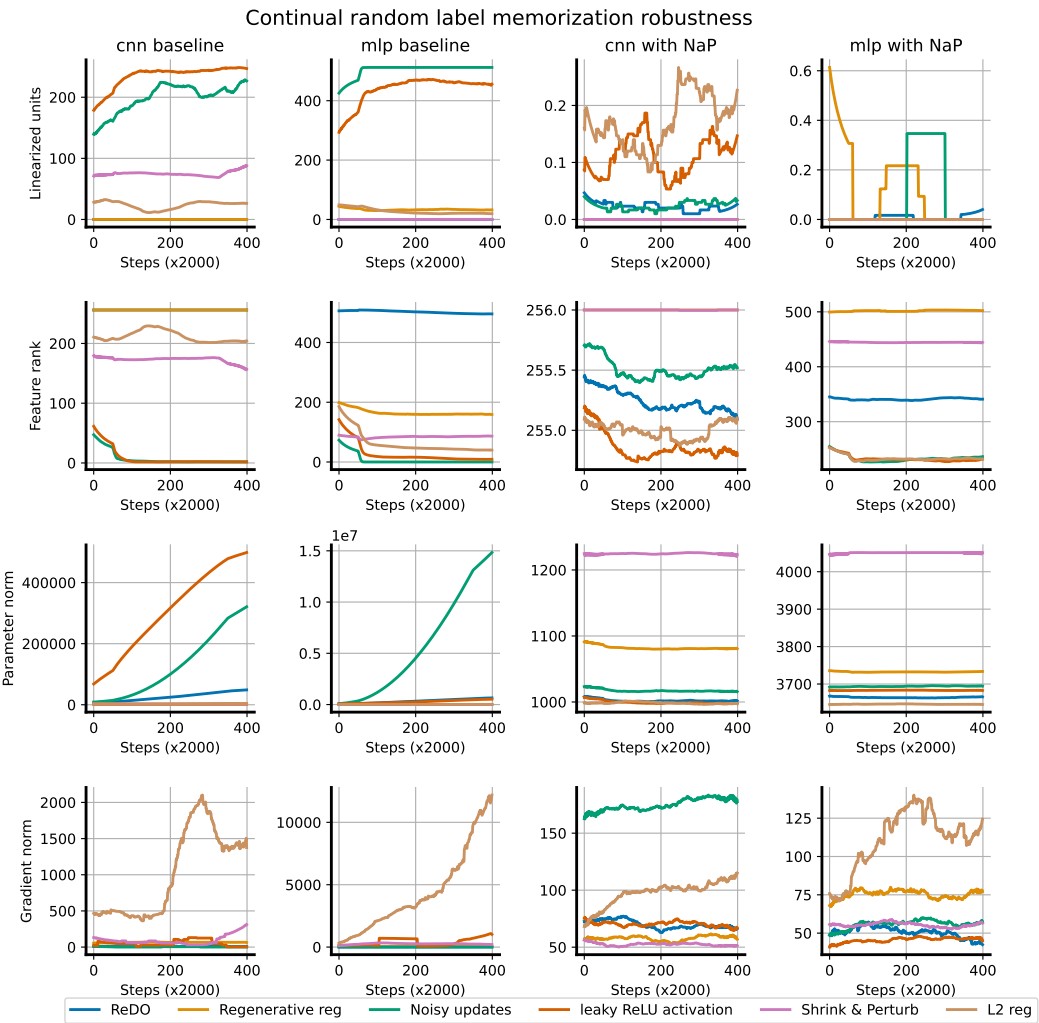

**Figure 8:** We plot a variety of additional network statistics in the continual CIFAR-10 experiment shown in Figure 5

the latter point in Figure 9, where we train a small transformer model on a dataset of the form $s_p \oplus s_s$, where $s_p$ is a prefix string of length 100 and $s_s$ is a suffix string of length 50 which is a contiguous subset of the prefix string. We use a fixed dataset, such that in theory the network could memorize all strings, though we use a sufficiently large dataset that this is not possible to achieve within the training budget. We train four protocols using next-token prediction: with and without weight projection, and with and without normalization (the networks are small enough that normalization is not critical for training stability). We evaluate accuracy on the final 48 tokens of $s_s$ as training progresses, and observe that all networks very quickly learn to identify the starting point of the suffix string and copy the relevant subset of the prefix. We observe that normalization accelerates learning of both the retrieval component of the accuracy and the memorization component of the accuracy, and that the weight projection step in fact also accelerates this process.

In Figure 10, we see that normalization and weight projection can also help to improve robustness in a nonstationary random string memorization task, a sequence-modelling analogue of random label memorization in CIFAR-10. We observe that in this case, allowing a learnable scale to evolve unregularized can somewhat slow down learning compared to omitting this parameter, and that weight projection recovers similar dynamics to learning with a large weight decay factor.

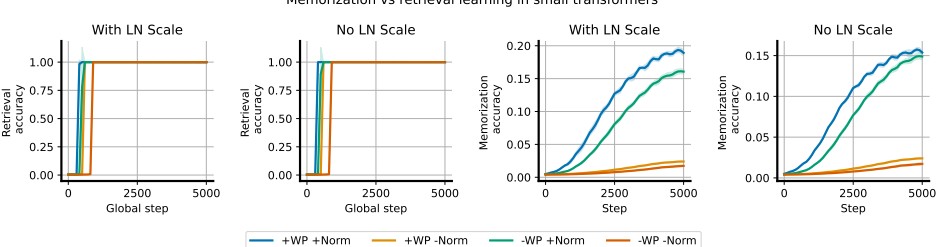

**Figure 9:** Demonstration that applying normalization and removing the learnable scale parameter does not prevent the network from learning to copy previously-observed subsequences.

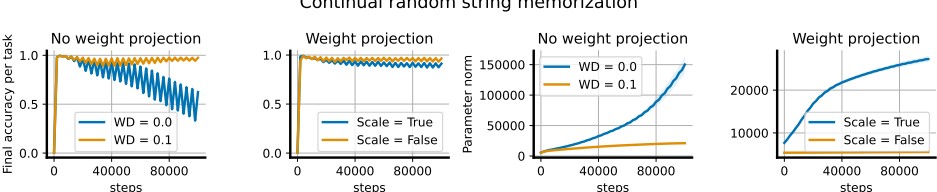

**Figure 10:** Random string memorization: unregularized networks exhibit plasticity loss when trained to memorize a sequence of random strings, while weight projection and weight decay improve robustness to this nonstationarity.

## C.4 ReLU revival experiments

Many previous works have noted that adaptive optimizers are particularly damaging to network plasticity [Dohare et al., 2021, Lyle et al., 2023, Dohare et al., 2023]. The primary mechanism underlying this is due to the sudden distribution shift in gradient moments due to changes in the learning objective – when the gradient norm increases, adaptive optimizers are slow to catch up and can take enormous update steps when this occurs. Layer normalization mitigates this effect due to two facts: first, the gradients of negative preactivations are nonzero, and second, all nonzero gradients are treated essentially the same by adaptive optimizers (up to $\epsilon$). As a result, networks with layer norm can still perform significant updates to parameters feeding into 'dead' units, meaning that these networks have a good chance of turning on again later.

We illustrate this with a simple experimental setting, where we model optimizer updates with isotropic Gaussian-distributed gradient signals and perform a (truncated at zero) random walk. Formally we look at the time evolution of the system:

$$\mathbf{v}_t = \mathbf{v}_{t-1} + \max(\mathbf{v}, \mathbf{0}) \odot \mathbf{z}_t, \quad \mathbf{z}_t \sim \mathcal{N}(0, I) \tag{30}$$

to model the evolution of features under a gradient descent trajectory. To simulate the steps taken by an adaptive optimizer like RMSProp or Adam, where updates to each parameter have fixed norm, we modify this process slightly as follows:

$$\mathbf{v}_t = \mathbf{v}_{t-1} + \text{sign}\left(\max(\mathbf{v}, \mathbf{0}) \odot \mathbf{z}_t\right), \quad \mathbf{z}_t \sim \mathcal{N}(0, I) . \tag{31}$$

To simulate layer normalization, we compute the dot product between $\mathbf{z}_t$ and the Jacobian $\nabla_{\mathbf{v}} \frac{\mathbf{v}}{\|\mathbf{v}\|}$ to simulate gradient descent:

$$\mathbf{v}_t = \mathbf{v}_{t-1} + \mathbf{z}_t^\top \nabla_{\mathbf{v}} \max\left(\frac{\mathbf{v}}{\|\mathbf{v}\|}, \mathbf{0}\right), \quad \mathbf{z}_t \sim \mathcal{N}(0, I) \tag{32}$$

and analogously compute the sign of this update to model RMSProp-style optimizers:

$$\mathbf{v}_t = \mathbf{v}_{t-1} + \text{sign}\left(\mathbf{z}_t^\top \nabla_{\mathbf{v}} \max\left(\frac{\mathbf{v}}{\|\mathbf{v}\|}, \mathbf{0}\right)\right), \quad \mathbf{z}_t \sim \mathcal{N}(0, I) \tag{33}$$

In Figure 11 we simulate each of these processes for 1000 steps, and track the number of negative (i.e. 'dead') indices. We observe that layer normalization doesn't avoid dead units in GD, but does reduce

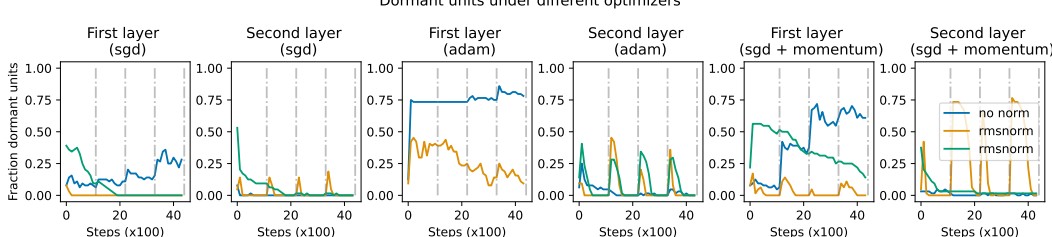

**Figure 11:** Simple MLP model with dead unit recovery after sudden changes in the classification task. We

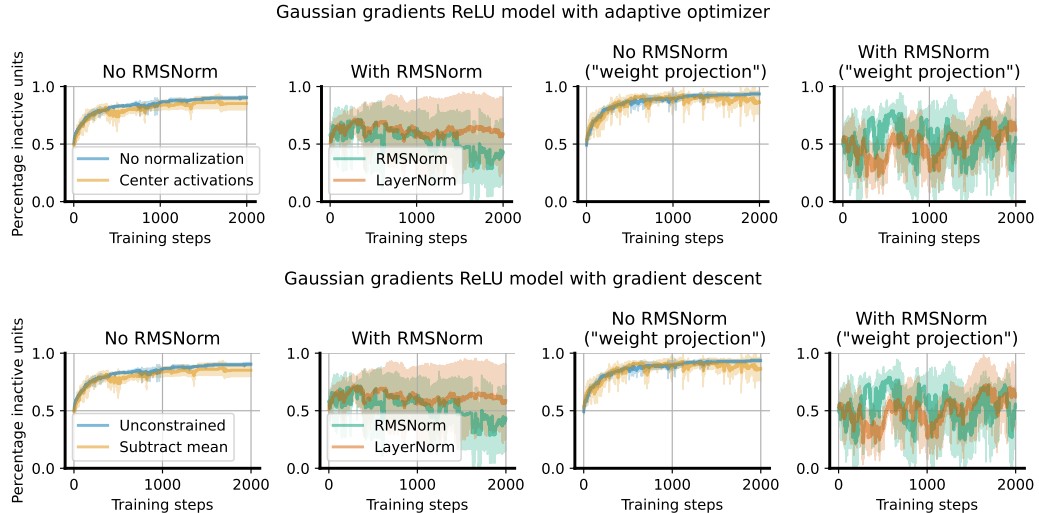

**Figure 12:** Accumulation of gradients in a random walk model: backpropagated 'gradients' are isotropic random Gaussian vectors and updates are computed by taking the product of these vectors with the layer jacobian. We see that the centering transform actually does relatively little to reduce the risk of dead units, and can in fact produce a 'winner-take-all' effect wherein one large

the rate at which they accumulate. We also observe that layer normalization does avoid monotonic increases in the number of dead units in a model of RMSProp. Intuitively, this makes sense: rather than freezing once they reach a negative value, parameters continue updating once they become negative with equally large steps as they did before, making escape from the dead zone more probable. One visualization of a few trajectories in each setting is shown in Figure 12.

## C.5 Stationary supervised benchmarks

We provide the results with standard deviations in Table 2 and Table 3.

|  | **CIFAR-10** | **ImageNet-1k** |
|---|---|---|
| NaP | **94.64 (0.12)** | 77.26 (0.04) |
| Baseline | **94.65 (0.08)** | 77.08 (0.11) |
| Norm only | 94.47 (0.18) | **77.45 (0.08)** |

**Table 2:** Top-1 prediction accuracy on the test sets of CIFAR-10 and ImageNet-1k. Numbers in parentheses are standard deviations.

|          | C4          | Pile        | WikiText    | Lambada     | SIQA        | PIQA        |
|----------|-------------|-------------|-------------|-------------|-------------|-------------|
| NaP      | **45.7 (0.0)** | **47.9 (0.1)** | **45.4 (0.1)** | **56.6 (0.4)** | **44.2 (0.2)** | **68.8 (0.7)** |
| Baseline | 44.8 (0.0)  | 47.4 (0.1)  | 44.2 (0.0)  | 54.1 (0.2)  | 43.5 (0.6)  | 67.3 (0.2)  |
| Norm only | 44.9 (0.0) | 47.6 (0.1)  | 44.3 (0.0)  | 53.6 (0.3)  | 43.8 (0.6)  | 67.1 (0.4)  |

**Table 3:** Per-token accuracy of a 400M transformer model pretrained on the C4 dataset, evaluated on a variety of language benchmarks. Numbers in parentheses are standard deviations.

# D  A note on scale and offset parameters

One loose end from our presentation of NaP is what to do with the learnable scale and offset terms, which are not by default projected and so may drift from their initial values. Most supervised tasks are too short for this drift to present problems, and adding layer-specific regularization or normalization adds additional engineering overhead to an experiment. However, in deep RL or in the synthetic continual tasks we present in Figure 5, this is a real concern. In the case of homogeneous activations such as ReLU, the scale and offset parameters can be viewed identically to the weight and bias terms and normalized accordingly, noting that now all that matters is the relative ratio of the scale and the offset. To account for this, we propose to treat the joint set $\sigma, \mu$ as a single parameter to be normalized. This resolves the issues involved with normalizing a parameter to an initial value of zero, and can be shown not to change the network output (see Appendix A.8 for a derivation of this fact). With non-homogeneous nonlinearities, however, this property will not hold, and we suggest in the general case to use mild weight decay towards a the initial values of 1 and 0 for the scale and offset terms respectively. These two approaches can be summarized in the following two update rules:

$$\mathcal{U}_{\text{norm}}(\sigma, \mu) = \frac{(\sigma, \mu)}{\sqrt{\|\sigma\|^2 + \|\mu\|^2}} \quad \text{and} \quad \mathcal{U}_{\text{decay}}^{\alpha}(\sigma, \mu) = (\alpha\sigma + (1 - \alpha)\mathbb{1}, \alpha\mu) . \quad (34)$$

Most of our evaluations on single tasks do not use any regularization or projection of the scale and offset parameters, but we do include a regularization-based approach in our evaluations on the sequential ALE in Section 5. In general, if it did not appear that the scale/offset drift was causing problems, we did not introduce additional complexity by adding regularization or normalization. Indeed, in many cases a simpler solution was to omit these parameters entirely; for example we observed that removing offsets was beneficial in several, though not all, games in the Arcade Learning Environment.

# E  Additional RL Experiments

## E.1  Actor-critic algorithms

We additionally investigate the effect of NaP on actor-critic methods, using the Brax implementations of SAC and PPO to evaluate the effects of the different design choices which contributed to the best-performing Rainbow agent in continuous control domains.

**Experiment details:** we base our experiments on the Brax [Freeman et al., 2021], using the implementation available at https://github.com/google/brax. We use networks of depth 4 with width 1024. When layer normalization is added to the architecture, we add it after each hidden layer's outputs in the actor and after all but the final two layer's outputs for the critic (we observe reduced performance when adding normalization to the penultimate layer as well, which we conjecture is due to the importance of euclidean distance in the final layer outputs in the continuous control tasks). All normalization layers have learnable scale and bias terms, which are not regularized or normalized in our experiments. We used either a constant learning rate schedule, or a linear schedule which decayed over 90% of the training budget to a final value of 1e-6. For each agent we considered two learning rates (1e-3 and 1e-4) and plotted whichever achieved the highest final performance.

**Effect of normalization and learning rate decay on performance:** we observe an environment-dependent ranking of the relative benefits from different design choices in Figure 13. In ant (leftmost figure), the linear learning rate schedule accounted for most of the improvement experienced by the full NaP approach. In combination with learning rate decay, layer normalization also tended to induce

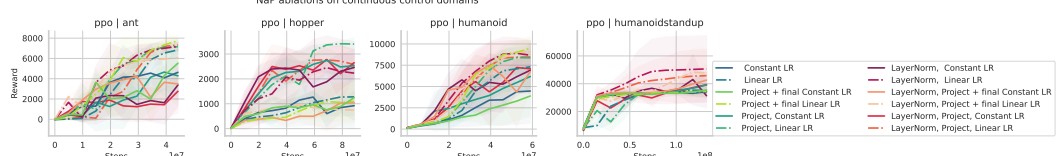

**Figure 13:** Ablations on learning rate schedule, weight projection, and layer normalization in PPO agents trained on mujoco environments.

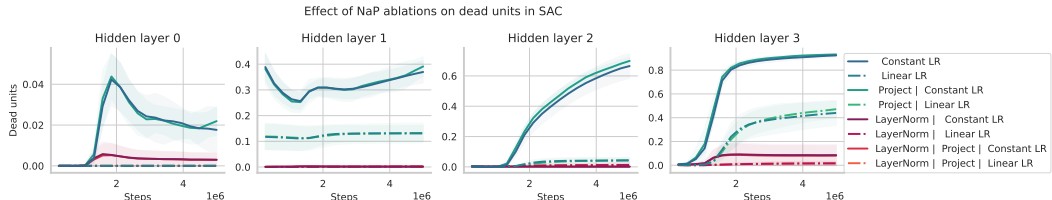

**Figure 14:** A closer look at the effect of NaP on dead units in a SAC agent trained on the Ant environment.

some improvement whether or not weight projection was used, but the effect size of this was much smaller than was observed in Atari. Weight projection had relatively little effect on performance, though the combination of layer norm, projection, and linear learning rate decay was consistently among the top-performing methods across environments.

**Effect of normalization and learning rate decay on dead units:** in Figure 14, track the accumulation of dead units across layers in the network of a SAC agent. We observe that weight projection has virtually no effect on dead units, with linear learning rate decay slowing the accumulation but layer normalization providing the greatest benefit, a finding consistent with observations from prior works.

## E.2 Detailed learning curves on Atari

For completeness, we include learning curves for all Rainbow variants in Figure 15.

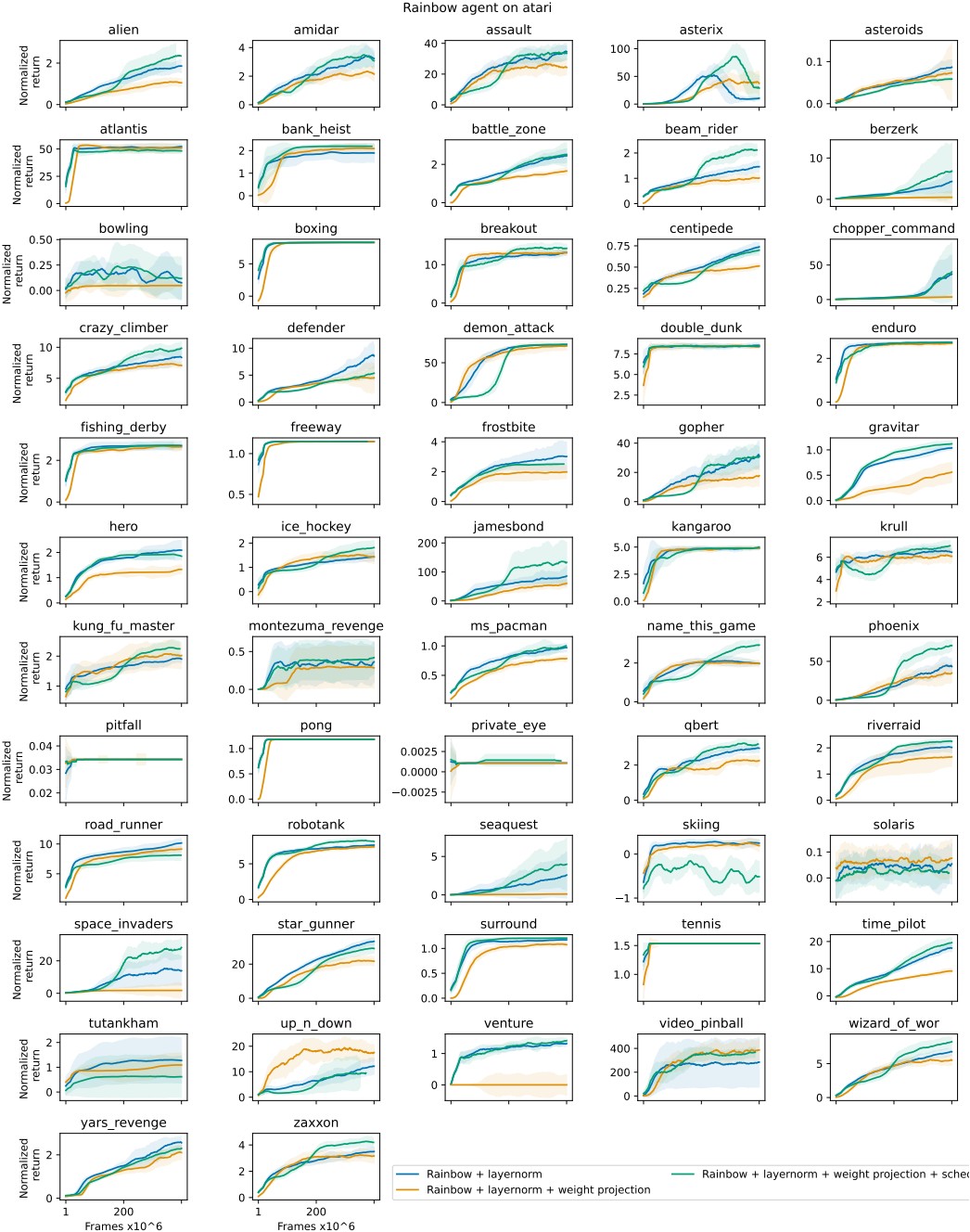

**Figure 15:** Rainbow agents on atari.

