# OpenReview forum: "Normalization and effective learning rates in reinforcement learning"
_NeurIPS.cc/2024/Conference — NeurIPS 2024 poster_

### Official Review · Reviewer_AEQp · 2024-06-27

**Soundness:** 3
**Presentation:** 3
**Contribution:** 3
**Rating:** 6
**Confidence:** 3

**Summary:**

This work attempts to improve the optimization for deep reinforcement learning by inserting additional layer norms into the architecture and performing weight projection steps that constrain the magnitude of the matrix weights. The paper discusses how the implicit effective learning rate schedule resulting from the growth of weight norms in standard optimization (without weight decay) affects reinforcement learning. The authors perform experiments across several domains, including non-stationary learning and standard deep reinforcement learning, generally obtaining improvements with their method.

**Strengths:**

+ The paper is well written overall with relatively clear figures and sufficient details (hyperparameters etc) to allow reproduction
+ The topic and proposed method are of relevance to the community
+ The proposed NaP method demonstrates good performance across varied experiments
+ The method is seemingly novel to the reinforcement field (but this not my background)
+ Variance quantification for at least some experiments

**Weaknesses:**

-  The NaP method seems to be closely related to various deep learning optimization methods that are not discussed in the paper (see details in question section). There should ideally be a comparison or at the very least a discussion of similar methods.
- The experimental procedure could be slightly more rigorous / complete in some places (see details in question section)
- Some parts of the paper are a bit unclear (details below)

**Questions:**

### Related work
Controlling some notion of an effective learning rate is not new to the broader field of deep learning. There are various optimizers that attempt to do this including LARS [1], LAMB [2], Nero [3], RVs [4]. Over time weight decay has also been shown to modulate the effective learning rate, bringing it towards a specific value, especially for SGD [5] and AdamW [4] (which should result in a very similar effect to projecting the weights, see discussion in [4]). It would be interesting to see a comparison or at least a discussion of these related works and techniques. Nero, RVs and the forced weight normalization of [6] use weight projections that should work very similarly to the proposed method (although on a finer granularity).

These optimizers and weight decay in general would probably also need a learning rate schedule similar to NaP. I find it weird that this is supposedly not standard practice in DeepRL. Even for a stationary distribution the learning rate needs to be decreased with a schedule to obtain good results when weight decay is used on top of stochastic optimization. This can be seen e.g. in the original AdamW work [7], where the optimal weight decay for standard CIFAR ResNet training is zero when a fixed learning rate is used but not with a cosine schedule.

* [1]: You, Yang, Igor Gitman, and Boris Ginsburg. "Large batch training of convolutional networks." arXiv preprint arXiv:1708.03888 (2017).
* [2]: You, Yang, et al. "Large batch optimization for deep learning: Training bert in 76 minutes." arXiv preprint arXiv:1904.00962 (2019).
* [3]: Liu, Yang, Jeremy Bernstein, Markus Meister, and Yisong Yue. "Learning by turning: Neural architecture aware optimisation." In International Conference on Machine Learning, pp. 6748-6758. PMLR, 2021.
* [4]: Kosson, Atli, Bettina Messmer, and Martin Jaggi. "Rotational equilibrium: How weight decay balances learning across neural networks." arXiv preprint arXiv:2305.17212 (2023).
* [5]: Wan, Ruosi, Zhanxing Zhu, Xiangyu Zhang, and Jian Sun. "Spherical motion dynamics: Learning dynamics of neural network with normalization, weight decay, and sgd." arXiv preprint arXiv:2006.08419 (2020).
* [6]: Karras, Tero, Miika Aittala, Jaakko Lehtinen, Janne Hellsten, Timo Aila, and Samuli Laine. "Analyzing and improving the training dynamics of diffusion models." In Proceedings of the IEEE/CVF Conference on Computer Vision and Pattern Recognition, pp. 24174-24184. 2024.
* [7]: Loshchilov, Ilya, and Frank Hutter. "Decoupled weight decay regularization." arXiv preprint arXiv:1711.05101 (2017).

### Experimental Procedure
The paper proposes two tricks, additional normalization and the weight projection. However the effects of each one individually are not evaluated very well, only their combination. For example in Figure 1, the effects of only weight projections without the additional normalization layers is not shown.

There is little comparison with simpler baselines. For example, if preventing dead ReLUs are the main advantage of layernorm, why not consider leaky ReLUs as a baseline?

The hyperparameter tuning is not very clear. In general the weight projection change is quite significant and should have the learning rate etc tuned separately from the baseline for a fair comparison.

### Clarity
Algorithm 1: This is quite unclear in the main body. Why are rho, mu, sigma not defined? Should these be arguments to the function along with W? Why is rho an argument but no value is provided when it is called? How do theta and theta prime relate to W? Is the semicolon subscript used to denote all layers or a subset of the matrix?

The whole description of the bias and gain handling is a bit unclear. I find it somewhat surprising that the “drift” causes problems in some of your settings. I wonder if this is due to the learning rate being too high for these parameters. One important property of weight decay is that it effectively defines a second learning rate for the bias and gain parameters by scaling the effective learning rate of the other weights (as shown / discussed in [4]). I wonder if this aspect is missing from NaP leading to some of these issues with gains and biases. Scaling the norm of the sphere you project onto might give you a similar effect.

**Minor suggestions and notes:**

L196: Algorithm reference is undefined

L224: This forward reference to Figure 2 is unclear since the relevant information for it has not been included and the figure caption only states “as described in text”.

L273: I disagree that the benefits of better conditioning should be independent of the effective learning rate in general. It is quite clear that if you set the effective learning rate sufficiently high or low you won’t learn anything, regardless of conditioning.

L278: This corresponds nicely with standard training, see e.g. example for ResNet AdamW mentioned above.

EQ14: This seems wrong. Is it supposed to be an inner product and a scalar zero?

L637: I don’t think this is correct, linear growth would result in something more like 1/x decay in the effective learning rate schedule, not something like 1-x which is typically meant by linear decay.

Figure 10: The caption is unclear or cut off.

EQ33: You don’t use the d term here like you do in the algorithm

**Limitations:**

Yes, no concerns here

---

> ### Author Rebuttal · Authors · 2024-08-06
>
> We thank the reviewer for their extremely helpful comments, in particular introducing us to a number of work on optimization dynamics in scale-invariant networks of which we were not aware. We address individual concerns below.
>
> **W1**: We thank the reviewer for the recommended citations, and will be sure to include them in our related work. Implementing these methods as baselines was unfortunately outside of the scope of the rebuttal period, but we have read the papers with great interest and plan on including them in our revisions. As the reviewer mentions they do not come from a reinforcement learning background, we want to emphasize that there is an extensive “graveyard” of tried-and-true supervised learning methods which have been unable to provide analogous benefits in reinforcement learning, including such basic methods as weight decay [3] and batch normalization [2].  We emphasize that translating an insight or technique from supervised learning to RL is often a highly nontrivial task (see e.g. [1] on the challenges of using momentum-based optimizers in RL), and hope that the reviewer takes this into account when assessing the significance of the contribution.
>
>
> ### Questions
>
> **Related work.** We thank the reviewer for the references, and will be sure to discuss them in the context of related work to the paper in our revisions. While it is an interesting direction for further work to explore these methods in RL, doing so was outside of scope for the rebuttal period. We reiterate that it is common for techniques standard in deep learning to fail to yield benefits in deep RL, and while it is likely possible that methods which are similar to our approach .
>
> **Ablating WP and LN**: We agree with the reviewer that the paper would benefit from including baselines where WP and LN are studied in isolation. An important point to emphasize is that without normalization layers, weight projection no longer carries th esemantics of constraining the effective learning rate. In our preliminary evaluations, we found weight projection alone without normalization to be a weak baseline and did not include it in our final results, however we will be sure to include this data in our revisions.
>
> **Why not leaky ReLUs**:
> We evaluate against a number of dead-unit-mitigating baselines in the experiments in Figure 4, including leaky ReLUs. While leaky ReLUs in particular provide some benefits, they do not typically completely mitigate plasticity loss (see e.g. [4] ).
> Further, while preventing dead ReLUs is one advantage of layernorm,  many other advantages have been noted as well [5], which we wished to benefit from.
>
> **Hyperparameter tuning**: For standard baselines (CIFAR/Imagenet, Rainbow) we use the default LR from the baseline method we compare against and set the LR schedule to end ~2 OOMs below this value (rounding to the nearest power of 10). We found that optimal LRs and schedules for NaP often involve slightly smaller initial learning rates and more aggressive decay than their counterparts (especially if no or weak weight decay or L2 regularization is used) but that it is typically more robust to the particular starting/ending values of the learning rate when using a schedule compared to a fixed value.
>
> **Algorithm 1 notation**: We thank the reviewer for highlighting this, and will clarify the notation for Algorithm 1 in our revisions to the paper. Specifically:
> - $\rho_l$ is the weight scale for a specific layer. We set this to be equal to its norm at initialization
> - $\mu_l$ and $\sigma_l$ are the (possibly vacuous) layernorm offset and scale terms respectively, which are learned parameters in the network. To make this clearer and to address another reviewer’s concern about the absence of an explicit definition of LayerNorm, we will include these terms in our LayerNorm expression which will be added to Section 3.
>
> **description of the bias and gain handling**: In most settings involving fewer than O(10^8) optimizer steps (i.e. all of the supervised learning benchmarks and single-task RL), we did not notice a significant effect from the choice of strategy for dealing with the bias and gain, and found even in the sequential ALE that the obvious solution of mild weight decay worked out of the box. We therefore did not devote much space in the paper to discuss how best to deal with these parameters as this choice does not appear to be practically significant. We will be sure to emphasize this more in our revisions.
>
> **Learning rate for the bias and gain parameters**
> We thank the reviewer for the pointer to the reference. We agree that in principle a similar phenomenon should be at play here, with NaP behaving similarly to the RV-AdamW method in [4 (Reviewer’s reference)]. While we unfortunately did not log the relative values of the scale and offset parameters in our continual atari experiments, we do observe that the total norm of these parameters does noticeably increase over the course of training if they are allowed to evolve unimpeded (the total parameter norm grows from ~135 to ~330, and all of this growth is attributable solely to the scale/offset terms).
>
> [1] Correcting Momentum in Temporal Difference Learning. Emmanuel Bengio, Joelle Pineau, Doina Precup. https://arxiv.org/abs/2106.03955
>
> [2] Liu, Zhuang, et al. "Regularization Matters in Policy Optimization-An Empirical Study on Continuous Control." International Conference on Learning Representations.
>
> [3] Salimans, Tim, and Durk P. Kingma. "Weight normalization: A simple reparameterization to accelerate training of deep neural networks." Advances in neural information processing systems 29 (2016).
>
> [4]  Lyle, Clare, et al. "Disentangling the causes of plasticity loss in neural networks." Third Conference on Lifelong Learning Agents (2024).
>
> [5] Nauman, Michal, et al. "Overestimation, Overfitting, and Plasticity in Actor-Critic: the Bitter Lesson of Reinforcement Learning." Forty-first International Conference on Machine Learning.

---

> > ### Comment · Reviewer_AEQp · 2024-08-09
> >
> > I thank the authors for their detailed response and clarifications. I agree that the core contribution of demonstrating these methods in DRL is valuable and non-trivial. I will raise my review score to 6.
> >
> > A few minor notes on the rebuttal:
> >
> > > An important point to emphasize is that without normalization layers, weight projection no longer carries the semantics of constraining the effective learning rate.
> >
> > I am not sure this actually matters that much, especially with normalized optimizers like Adam. Aside from the magnitude on the forward pass (assuming normalization of the activations and not the weights), the only aspect that changes is the removal of the component of the gradient parallel to the weights. Unless this component is very large, the resulting optimization dynamics (measured via relative or angular updates) will remain very similar.
> >
> > > In our preliminary evaluations, we found weight projection alone without normalization to be a weak baseline and did not include it in our final results, however we will be sure to include this data in our revisions.
> >
> > Just in case you are not familiar with this, there is a line of work that shows that normalization layers serve an important role in the signal propagation of networks with residual connections, see e.g. https://arxiv.org/abs/2002.10444. When removing activation normalization layers it is important to preserve these signal propagation dynamics with methods like those described in e.g. https://arxiv.org/abs/2102.06171 (the initialization / downweighting of the residual branches mostly, the weight standardization is unlikely to matter with Adam + projection in my opinion). I hope you account for this in your baselines, but including the results would be interesting either way.

---

> > > ### Author Response · Authors · 2024-08-12
> > >
> > > We thank the reviewer for their consideration of our rebuttal and for the additional references. Regarding the first point: we now realize that we misunderstood the original point in the review, as in many of the architectures we evaluated (particularly those used in value-based deep reinforcement learning) no normalization layers are included by default, and so removing the normalization introduced by the method would destroy the network's scale-invariance. However, we agree that isolating the effect of the *additional* layers on e.g. transformers or ResNets would be a useful baseline to include and plan to do so in our revisions. Regarding the second point: while exploring the interaction between residual connections, layer normalization, signal propagation and effective learning rates was outside the scope of our submission, we are keen to explore it in future work and agree that the two papers mentioned are very relevant to this direction.

---

### Official Review · Reviewer_vdAM · 2024-07-08

**Soundness:** 3
**Presentation:** 2
**Contribution:** 3
**Rating:** 6
**Confidence:** 3

**Summary:**

This paper explores the use of normalization layers in deep reinforcement learning and continual learning, as well as their impact on the effective learning rate. Although normalization layers offer a variety of benefits in stabilizing optimization and improving the loss landscape, they also introduce a significant side effect: the growth of the network parameter norm is equivalent to the decay of the effective learning rate. In continual learning environments, the implicit decay of the learning rate due to the increase in parameter norm may drop too quickly relative to the timescale of the learning problem, which is detrimental.

Therefore, this paper proposes a new method called Normalize-and-Project (NaP). The NaP method consists of two main steps: inserting normalization layers in the network architecture before non-linear activation functions; and regularly projecting the network's weights onto a fixed norm radius during the training process, along with corresponding updates to the learning rates for each layer in the optimization process. This approach ensures that the effective learning rate remains constant throughout the training.

Furthermore, this paper validates the effectiveness of the NaP method through a series of experiments, demonstrating its potential to enhance performance and robustness across different learning environments and settings.

**Strengths:**

Firstly, this paper addresses a significant issue within the machine learning community, namely, the plasticity of neural network learning. It offers novel and intriguing insights into the widely-used layer normalization technique, revealing its advantages in controlling the growth of parameter norms and its disadvantages in continual learning settings. Thus, the paper provides innovative insights.

Secondly, the paper presents a concise and effective method that builds upon the commonly used layer normalization by incorporating a parameter norm projection step, making it more suitable for continual learning scenarios. Consequently, the method proposed in this paper has universality and practicality.

Lastly, through a series of experiments, the paper demonstrates that the NaP method does not affect learning in static tasks and has distinct advantages in continual supervised learning and reinforcement learning tasks.

Additionally, the paper is supported by theoretical analysis and ablation study evidence, which solidify and complete the arguments presented in the paper.

**Weaknesses:**

1. This paper lacks a unified mathematical expression framework; the mathematical notation for many theoretical conclusions is disjointed and abrupt, increasing the cognitive cost for readers. For instance, throughout the paper, the specific definition of layer normalization is not explicitly provided, which leaves me somewhat confused about whether normalization is applied to the parameters of each layer itself or to the inputs of the layer. Moreover, in Definition 1, parameters are denoted by \(\theta_t\), but in Proposition 1, a new symbol \(h\) is introduced without clarifying its relationship to \(\theta_t\); furthermore, in Algorithm 1, the left column only uses WeightProject(\(W_l\)), while the right column defines WeightProject(\(W_l\), \(\rho_l\)) and employs an undefined function \(len\). Therefore, I suggest the authors carefully review the mathematical language they have used.

2. Although the title of this paper indicates a focus on reinforcement learning, more than half of the experiments in the paper are based on supervised learning or artificial experiments, and the discussion and theory in this paper do not seem to be specific to reinforcement learning. Therefore, the authors need to discuss the strong coupling relationship of this scheme with reinforcement learning. In addition, there are some logically inconsistent aspects in this paper. On one hand, the paper argues that the decay of the learning rate is inappropriate in the supervised learning setting, which supports the design concept of NaP; on the other hand, in the reinforcement learning setting, learning rate decay is needed, but NaP cannot provide an independent solution, that is, it still requires some learning rate scheduling strategies. Therefore, I am more skeptical about the universality of the NaP method for the RL field.

**Questions:**

1. Please clarify the standard mathematical definition of layer normalization and its relationship with the notation used in the various mathematical theories presented in this paper. Specifically, it would be helpful to understand how the notation for layer normalization relates to the symbols used in Definition 1， Proposition 1 and Algorithm 1, and to have a clear explanation of the transition from \(\theta_t\) to \(h\) in the context of the paper's mathematical framework.

2. Elucidate the strong coupling relationship between NaP and reinforcement learning. Particularly, please supplement the performance of NaP on typical continuous tasks such as Gym Mujoco, because on these types of tasks, a decaying learning rate schedule design is often not required. This could potentially better demonstrate the advantages of NaP, as it may allow for a more consistent learning rate that could be beneficial in environments where the learning problem does not necessitate a reduction in learning rate over time.

**Limitations:**

The authors have not discussed the potential negative impacts of this paper, but I believe this work indeed has no significant negative societal impacts.

---

> ### Author Rebuttal · Authors · 2024-08-06
>
> We thank the reviewer for their careful engagement with the manuscript and for their helpful comments. We will ensure to take these into account in our revisions.
>
> **Mathematical notation.** We thank the reviewer for highlighting this. We will be sure to review the mathematical notation and improve it in our updates based on these suggestions. We had left out a formal definition of layer normalization because of its widespread use, but can include LN explicitly in the same section where we define RMS-norm, and will also include the following clarifications:
> -  h refers to a hidden unit’s pre-activation
> - $\rho_l$ refers to the reference norm to which layer l is projected.
> - Len is standard length function which returns the length of a tuple/vector as in python.
>
> **Strong coupling relationship of this scheme with reinforcement learning.**  While we agree with the reviewer that NaP can be applied outside of reinforcement learning problems, we do not view the generality of our approach as a weakness but rather as a strength: NaP is not only a highly effective tool for understanding and improving reinforcement learning algorithms, but also more broadly applicable to other nonstationary learning problems. We emphasize RL in the title because it is both the regime where NaP is most beneficial, and also the regime where NaP offers the most insight as a diagnostic tool. In particular, NaP yields significant performance improvements in both single- and multi-task variants of Atari, and further demonstrates the surprising and previously unknown importance of ELR decay in this regime when used as a diagnostic tool. Figure 6 in the Appendix offers particularly intriguing hints at the possibility that learning certain components of a value function require dropping the learning rate below some critical threshold. We will be sure to emphasize this point in our revisions to make the relationship between NaP and reinforcement learning clearer.
>
> **... the paper argues that the decay of the learning rate is inappropriate in the supervised learning setting, which supports the design concept of NaP.** We emphasize that we **do not** claim that learning rate decay is always bad for network training dynamics – rather, we argue that *unintended* ELR decay due to parameter norm growth can slow down learning in some cases, particularly in long training runs such as those used in continual learning benchmarks. Some learning rate decay can be appropriate for supervised learning, and is indeed a standard part of neural network training. Our emphasis is that for long and/or nonstationary training problems, parameter norm growth can induce *excessive* decay that slows down learning if it is not controlled for in some way. We will endeavour to make this point clearer in our revisions, and thank the reviewer for highlighting this ambiguity in our original text.
>
> **on the other hand, in the reinforcement learning setting, learning rate decay is needed, but NaP cannot provide an independent solution, that is, it still requires some learning rate scheduling strategies.** As our previous point has hopefully clarified, the main message of this paper is not that ELR decay is bad, but that depending on growth in the parameter norm to decay the ELR is sub-optimal compared to tuning the schedule explicitly. The strength of NaP is that it allows the learning rate schedule to be set based on prior knowledge of the problem, rather than depending on incidental growth in the parameter norm. In the case of single-task RL, for example, the natural parameter norm growth is too mild, and we find that a more aggressive schedule is beneficial. By contrast, in the sequential atari benchmark, the natural decay is too extreme and a cyclic schedule that periodically re-warms the learning rate is preferrable instead, an intuitive solution given the cyclic nature of the problem.
>
> **“Skeptical … universality of the NaP method for the RL field.”** We find that the basic principle of “start at the default learning rate used by the baseline, then decay by 2 orders of magnitude” is a relatively robust recipe for not just Atari but also the continuous control tasks “Ant” and “Humanoid” with PPO agents (see our “General Comment” for numerical details from these experiments). We believe that even better LR schedules which depend on properties like the signal-to-noise ratio in the gradients and the local curvature of the loss landscape can likely provide further benefits, and are an exciting direction for future work.

---

### Official Review · Reviewer_KJMZ · 2024-07-10

**Soundness:** 3
**Presentation:** 3
**Contribution:** 4
**Rating:** 7
**Confidence:** 3

**Summary:**

Normalization layers improve various aspects of deep RL and continual learning, such as loss landscape conditioning and reducing overestimation bias. However, normalization can inadvertently decrease the effective learning rate as network parameters grow. This effect is problematic in continual learning where the learning rate can decay too quickly. This work proposes a re-parameterization method called Normalize-and-Project (NaP) to maintain a consistent effective learning rate throughout training, improving performance in both stationary and nonstationary environments.

 - Normalization layers stabilize optimization by conditioning the loss landscape and mitigating overestimation bias.
They create a scale-invariance in the network parameters, leading to a decline in the effective learning rate as the parameter norm increases.
Normalize-and-Project (NaP) - NaP is a protocol combining normalization layers with weight projection to keep the effective learning rate constant. It involves inserting normalization layers before nonlinearities and periodically rescaling the network’s weights to maintain a fixed norm.
-The paper explores how normalization layers affect a network’s plasticity and introduce the concept of effective learning rates.
It shows that normalization can lead to implicit learning rate decay, which can be beneficial or harmful depending on the learning context.
 -The paper evaluates NaP on various architectures and datasets, including RL tasks and benchmarks like CIFAR-10 and ImageNet.
NaP demonstrates improved robustness to nonstationarity and maintains performance in stationary settings.
-Loss of plasticity is a major barrier in RL and continual learning.Normalization layers help maintain plasticity by reducing parameter norm growth and stabilizing gradients.
-NaP can be easily integrated into existing architectures like ResNets and transformers.It provides a framework for better understanding and managing learning rate schedules in nonstationary problems.

The paper validates NaP through experiments on synthetic tasks and large-scale benchmarks, showing consistent improvements in performance and stability.It highlights the importance of controlling parameter norms and maintaining effective learning rates to mitigate plasticity loss.

**Strengths:**

Originality
Innovative Method: The introduction of Normalize-and-Project (NaP) is a novel approach that addresses the challenge of effective learning rate decay in deep reinforcement learning and continual learning.
Creative Combinations: Combining normalization with weight projection to maintain a consistent learning rate is both original and practical.
New Application: Applying these concepts to nonstationary reinforcement learning settings is a novel contribution.

Quality
Theoretical Rigor: The paper provides some theoretical insights into the relationship between parameter norms, learning rates, and network plasticity.
Empirical Validation: Robust experiments across various architectures and datasets convincingly demonstrate NaP's effectiveness.
Reproducibility: Clear methodology and detailed descriptions ensure that the work can be reproduced and built upon.

Clarity
Clear Writing: The paper is well-written and logically structured, making it easy to follow the arguments and results.
Good Coverage: Background and related work sections provide context and help readers of all levels grasp the contributions.

Significance
Broad Impact: NaP has the potential to significantly impact deep reinforcement learning and continual learning by addressing a fundamental challenge.
Practical Applicability: The method can be easily integrated into existing architectures, enhancing its real-world relevance.
Advancing Understanding: Theoretical insights advance the field's understanding of learning rates and network plasticity, suggesting new research directions.
Overall Assessment
The paper combines originality with practical impact, offering a novel solution to a key challenge in reinforcement learning and continual learning. Its good theoretical and empirical work, clear exposition, and significant contributions make it a valuable addition to the field.

**Weaknesses:**

Lack of Theoretical RL Analysis: The paper primarily focuses on the impact of normalization on network parameters and effective learning rates without delving into the theoretical implications from a reinforcement learning (RL) perspective, particularly regarding policy learning.
Potentially include a section discussing how the NaP method affects the learning of policies in RL. Explain how the steps in network parameter space translate to changes in policy space and the potential impact on policy optimization. This addition would provide a more comprehensive theoretical foundation and align the method's analysis with RL-specific objectives.

Experiments
Add a continuous control tasks or more complex environments, as in environment with more actions,  beyond the Atari suite, to add evidence to the generality and robustness of NaP across different RL scenarios.

**Questions:**

No questions.

**Limitations:**

Yes.

---

> ### Author Rebuttal · Authors · 2024-08-06
>
> We thank the reviewer for their engagement with the paper, and for their constructive comments. We address individual concerns below.
>
> **Lack of theoretical RL analysis:** We agree with the reviewer that translating insights on the optimization and trainability of neural networks to policy optimization is an interesting area for future work. Given the relatively nascent state of the field’s understanding of the relationship between optimization dynamics and policy learning, particularly in actor-critic settings (see e.g.[1]), we are not currently aware of an existing framework in which we could insert our method to study its impact on policy learning and would be interested in hearing if the reviewer had a particular flavour of result in mind. One result from our paper which could provide an interesting foundation for future investigation in this direction is Figure 6 in Appendix C, which shows that inflection points in the agent’s learning curve across different schedules tend to occur when the schedules reach a particular learning rate, suggesting that some environments require decaying the learning rate to a particular value for the network to be expressive enough for an optimization step to translate to an improved policy. Additionally, we are interested in exploring whether the optimal ELR for the actor and critic may differ in policy gradient algorithms.
>
> **Experiments.**  We have evaluated NaP on PPO agents trained with the Brax framework on Mujoco and have included these results in our general comment.
>
> [1]  Ilyas, Andrew, et al. "A Closer Look at Deep Policy Gradients." International Conference on Learning Representations.

---

### Official Review · Reviewer_6ZG9 · 2024-07-12

**Soundness:** 3
**Presentation:** 3
**Contribution:** 3
**Rating:** 7
**Confidence:** 4

**Summary:**

When training a neural network with layer normalization, an increase in the norm
of the parameters can lead to a lower effective learning rate. This paper makes
the observation that, when layer normalization is used, periodic projection is
enough to overcome this vanishing step-size. The idea is verified empirically,
and involves essentially no tuning due to the norm being set to initialization
and the periodicity of the projection does not significantly impact the
performance.



Decision: This paper provides a clear articulation of a relatively straightforward
observation. However, the implications of the observation seem wide ranging:
from non-stationary problems to reinforcement learning and even supervised
learning. Although the method is relatively simple and lacks "novelty", I think
this paper merits acceptance. It could be further improved by providing a more
nuanced investigation of the interplay between layer normalization and
normalize-and-project.

After rebuttal: I have updated my score to reflect the additional results in the shared reply and the discussion with authors (6 -> 7)

**Strengths:**

- The problem of reduced effective learning rate is clearly articulated. The solution is simple and straightforward, translating to improved performance across a wide range of problems: non-stationary supervised learning, reinforcement learning and even stationary learning tasks.
- The toy experiments presented before the main experiments are convincing. I particularly like the result with the coupled networks,

**Weaknesses:**

- Some of the arguments are ad-hoc, and rely on "folk knowledge" specific to deep learning. Layer normalization is indeed commonly used, and it is a good starting point, but understanding the interplay of layer-normalization and the normalize-and-project (in particular, to what degree it affects capacity) would provide more insight.
- The empirical results, while comprehensive, provide little further insight in addition to the toy examples presented earlier.

**Questions:**

- Section 1:  "We first show that when coupled with adaptive optimizers [..] saturated units still receive sufficient gradient signal to make non-trivial changes to their parameters"
  Is this claim at odds with the growing empirical evidence that loss of plasticity often occurs (without normalization layers)?

- Section 2.1: "learning dynamics can be well-characterized in the
  infinite-width limit [...], although in practice optimization dynamics diverge
  significantly from the infinite-width limit"

  What is the distinction between learning dynamics in the infinite-width limit and the optimization dynamics in the infinite width limit? I am not sure what the takeaway of this discussion should be.

- Section 2.1: "Plasticity loss can be further decomposed into two distinct components [Lee et al., 2023]"
  While I agree that loss of plasticity can be thought of as loss of trainability and loss of generalization, the provided reference discusses input and label plasticity. I do not see the distinction between trainability and generalization discussed in the paper.

- Section 2.2: "$\nabla f(c\theta) = \frac{1}{c} \nabla f(\theta)$"
  There is some ambiguity here in what the gradient operator is with respect to. For example, $\nabla_\theta f(c\theta)  = c \nabla_{c\theta} f(c\theta)$. If the parameter norm is large, then the parameter is not defined as the effectively normalized parameter but the "raw" parameter $c \theta$.
  The intuition provided is not quite as simple as the authors make it seem: it assumes that the perturbations are independent of the scale.

- Section 4.2: The schedule proposed seems completely ad-hoc and the Appendix does not describe this in sufficient detail. Could the authors comment on how "schedule misspecification" may impact performance? For example, does this Atari-based schedule harm performance on other RL tasks? If so, then the use of the schedule seems to use relatively priviledged knowledge for a continual learning algorithms (it requires running an algorithm on the entire continual learning problem to track the parameter growth)

- Section 5.1: "Further, we observe constant or increasing slopes in the online accuracy, suggesting that the difference between methods has more to do with their effect on within-task performance "

  Can you clarify how the slope of the online accuracy has to do within-task performance rather than loss of plasticity? It could be that the effect of NAP is redundant in conjunction with some of these methods. Regenerative regularization, for example, likely keeps the norm close to that of NAP.

- Section 5.2: The results on language tasks is interesting, because some degree of weight decay is usually used to train transformers (AdamW is a common choice, which your experiments also use). What effect does NAP have in conjuction with weight deacy? This is obviously related to my previous question, but NAP+AdamW seems redundant.

- Section 5.3: I find the use of knowledge of the game switch to be against the
  spirit of sequential ALE. The results are interesting, and impressive with
  this caveat. But the results would be much more impressive if evaluated
  without this type of human intervention.

### Minor:
- Section 3.3: There is a reference to an Algorithm -1 which does not point to anything.

**Limitations:**

See above

---

> ### Author Response · Authors · 2024-08-06
>
> We thank the reviewer for their detailed and helpful comments, and for their deep engagement with the paper. We address specific comments and questions individually as follows.
>
> **W1:** “folk knowledge” … “interplay between layer normalization and NaP”:
>
> We agree that further theoretical analysis into the dynamics of NaP is an exciting direction for further research. However, we emphasize that the theory explaining much of the “folk knowledge” referred to by the reviewer has been extensively developed in recent years, giving NaP a more rigorous grounding (see e.g. Martens et al. (2021)). Further, the study of optimization in scale-invariant networks and the resulting tying of the parameter norm and learning rate has been extensively developed by the works highlighted in Section 2.2, among others. These works show that the interaction between layer normalization and the parameter norm is precisely the effective learning rate. The interplay between the normalization and projection steps of NaP is therefore to keep the ELR exactly equal to the explicit learning rate schedule. Indeed, we motivate the weight projection step as a means of ensuring that the ELR does not *unexpectedly* shrink or grow over the course of training as a result of unintended changes in the parameter norm. We hope that future work may yield further insights into the implications of these works in the reinforcement learning context.
>
> Regarding capacity, we show in Proposition 2 (Appendix A.7) shows that NaP does not change a particular notion of network expressivity studied extensively by Poole et al.
>
> **W2:** "The empirical results [...] little further insight."
>
> While we agree with the reviewer that the primary function of our supervised learning evaluations is to simply verify that NaP does not harm performance on standard benchmarks, we note that our experiments in the deep reinforcement learning settings do provide additional nuance to our understanding of NaP. In particular, we find that contrary to folk wisdom, decay in the effective learning rate can indeed be beneficial in reinforcement learning. Further, our detailed learning curves in Figure 6 illustrate how upticks in performance frequently occur when the network passes through a specific range of learning rates, suggesting that different learning rates are needed to learn different skills within a task. These insights could not have been deduced from the toy experiments, and demonstrate that NaP can be used to gain new insights into well-studied problem settings.
>
> **[Q1]**
>
> There are a variety of reasons for loss of plasticity in neural networks. Some of these involve saturated units, while others are more complex and involve collapse of the features or starvation dynamics.  Proposition 1 notes a new mechanism by which layer normalization helps to protect against plasticity loss, but does not claim that this implies LN completely mitigates the plasticity loss phenomenon.
>
> **[Q2-4]** We thank the reviewer for highlighting these sources of ambiguity. We will rewrite [Q2] to clarify that we mean that infinite-width dynamics (optimization/training are equivalent in this sentence) are well-characterized theoretically but don’t accurately capture empirically-observed dynamics in finite-width settings. The “input plasticity” referred to by Lee et al. in [Q3] subsumes the “warm-starting effect”, where networks generalize worse after being trained on nonstationary inputs. We will clarify this in our revisions.  We can confirm that the gradient is w.r.t. the inputs of f in [Q4]. We will rewrite this statement to clarify that we are interested in computing the gradient of f w.r.t. Inputs $\theta$ evaluated at two particular values of $\theta$. I.e. $\nabla_\theta f(\theta) $ evaluated at $\theta_1$ vs $\nabla_\theta f(\theta)$ evaluated at ${\theta_2}$ where $\theta_1 = c \theta_2$ for some $c$.
>
> **[Q5]** please see “general comment” (choice of learning rate schedules/schedule misspecification).
>
> **[Q6]** We understand the reviewer’s confusion, as we did not have space in the paper to include per-task learning curves in Figure 4. Our comment refers to the fact that even in the first task, some strategies, in particular those which add noise to the training process, result in slower learning and `shallower’ learning curves. We will be sure to add these results to the supplementary material in our revisions and refer to them in this section of the paper.
>
> **[Q7] ** On projected parameters, adamw has no effect because its parameter scaling step is immediately undone by the projection step (though we might see slightly more complex behaviour with L2 regularization). However, because transformers have a number of nonlinear layers (in particular positional encodings) which are not invariant under parameter projection, some form of weight decay is likely helpful to improve stability in these layers where we can’t project.
>
> **[Q8]** please see "general comment" (continual atari schedule).

---

> ### Comment · Reviewer_6ZG9 · 2024-08-11
>
> The shared reply addresses many of my concerns, and I will be increasing my score. However, I have a few followups:
>
> - Re Q2-Q4: "The “input plasticity” referred to by Lee et al. in [Q3] subsumes the “warm-starting effect”, where networks generalize worse after being trained on nonstationary inputs."
>
>   It is true that the warm-starting paper (Ash and Adams, 2019) investigates loss of generalization. It is also true that the problem studied in that paper has changes in the input distribution. But I am struggling to see how input plasticity would "subsume" warm-starting. I think this would require further explanation to make this connection clear. One alternative is to separately reference papers that individually investigate loss of trainability and loss of generalization.
>
> - Re Q5/Q8 and general reply: I am pleased to see that the specifics of the learning rate schedule does not necessarily require knowledge of the parameter norm growth. I am also generally surprised by the Mujoco results you presented, especially the effectiveness of learning rate schedules. I understand LR schedules are somewhat common in supervised learning, I do not know how common they are in RL. I hope that the future revision of this paper reflect the surprising effectiveness of learning rate schedules in addition to the proposed method (or otherwise points to previous work that investigates LR schedules in RL).
>
> - Re Q7: Thanks, I did not realize that the positional encoding layers would be trainable. The standard transformer architecture introduced by Vaswani et al. (2017) used non-trainable positional encodings. Could you comment on the specifics of the positional encoding used?

---

> > ### Author Response · Authors · 2024-08-12
> >
> > We thank the reviewer for their consideration of our rebuttal. We appreciate the additional suggestions, which we believe will further improve the paper. Concretely:
> >
> > - [Q2-Q4]: We will take this point into account in our revisions and plan to clarify the sentence per the reviewer's suggestion by citing separate papers describing input plasticity and loss of generalization ability.
> >
> > - [Q5/Q8]: We are glad the reviewer is satisfied by our response. While it is likely that some RL papers have used non-constant learning rate schedules, a quick survey of papers which proposed popular RL methods (e.g. [PPO](https://arxiv.org/pdf/1707.06347), [DQN](https://arxiv.org/pdf/1312.5602), [SAC](https://arxiv.org/pdf/1801.01290), and [CURL](https://arxiv.org/pdf/2004.04136)) suggests that their use is quite rare, and has to the best of our knowledge not been studied systematically. The reviewer is correct that this lack of analysis starkly contrasts to the deep study of learning rate schedules in supervised learning problems. In this context, our findings on the benefits of learning rate schedules in deep RL are quite surprising and suggest that the learning rate schedule has been overlooked relative to other hyperparameters to the detriment of performance on benchmarks. We will be sure to emphasize this point in our revisions.
> >
> > - [Q7]: We use relative positional encodings as described in [1], which involves learnable weights, and will be sure to make this point clearer in our architecture description in our revisions.
> >
> > [1] Self-Attention with Relative Position Representations. Shaw et al., ACL 2018. https://arxiv.org/pdf/1803.02155

---

### Author Rebuttal · Authors · 2024-08-06

We thank the reviewers for their careful engagement with the work and thoughtful reviews. Reviewers generally agreed on the “wide ranging” implications of our findings [6ZG9], along with the “good theoretical and empirical work”[KJMZ], and “novel and intriguing insights”[vdAM], agreeing that while developing on a strong body of existing work in deep learning theory, our findings were “novel to the reinforcement [learning] field” and our method “demonstrates good performance across varied experiments”[AEQp].

We have addressed specific comments in individual rebuttals, and respond to shared concerns in this general note.
**Notation:** we clarify the following notation for Algorithm 1, which we will incorporate in our revisions:

**Choice of learning rate schedules:**
While any particular learning rate schedule may seem arbitrary, we emphasize that we chose each task's schedule with relatively little tuning -- in the case of supervised benchmarks we swept over number of training steps and initial learning rate and used the benchmark's default schedule. Our choices for RL were inspired by parameter norm growth curves, but the principles underlying them can be applied more broadly (and indeed, were applied to our Mujoco experiments that follow with relative ease *without* any prior knowledge of the baseline parameter norm growth).

**Sensitivity to schedule mis-specification:**
Just as using a too-large or too-small learning rate can result in poor performance in deep RL, we find analogously that using a schedule that decays too quickly, or which starts at too high of a value and decays too slowly, can hurt performance. In general, we find that schedule misspecification is slightly more forgiving than standard learning rate misspecification where the learning rate is held fixed throughout training. By letting training pass through a wider range of learning rates, we increase the probability that the network will spend at least some time in the optimal range for the given problem.

**Continual atari schedule:** The purpose of this experiment was to show that *there exists* a learning rate schedule under which we do not see loss of plasticity on the sequential ALE. We think further work exploring how to adaptively set the learning rate in response to environment nonstationarity is a particularly exciting direction to eliminate the need for this human intervention. In this particular setting, we expect that using a sufficiently good changepoint detection algorithm as a trigger for schedule resets would be sufficient to recover the performance of the handcrafted schedule.

**Mujoco experiments:** Multiple reviewers have requested evaluations on continuous control domains such as mujoco to complement our results in Atari. While we did not have sufficient time during the rebuttal period to conduct and hyperparameter-tune long-running experiments, we found that using the highly parallel brax library enabled sufficiently speedy iteration time that we could get reasonable baselines running fairly quickly, and provided a reasonable set of default hyperparameters. We have included results from training PPO a handful of popular continuous control benchmarks in the following table, in order to highlight the versatility of our method, based on the hyperparameters outlined in [this iPython notebook](https://github.com/google/brax/blob/main/notebooks/training.ipynb), using a 4-layer, 1024 width MLP as the network architecture for both actor and critic networks.

The results here suggest that while different design choices contribute more or less significantly to performance depending on the environment (for example, the ant environment benefits most from a learning rate schedule, whereas humanoid-stand benefits most from layer normalization), the general recipe of normalization + LR schedule consistently outperforms the standard baseline, and is more robust to varying the learning rate. We do not observe significant parameter norm growth in any of these environments due to the comparatively short training times relative to Atari (even in the longest-running task humanoid-stand, parameters increased by less than an order of magnitude). We also note that due to the large scale of the value function in these environments, using layer norm and weight projection can make it difficult for the critic network to accurately predict the value. We therefore found it beneficial in these domains to omit the normalization in the final layer of the critic network to allow it to scale its outputs to the target magnitude.

| PPO                                            | ant      | hopper   | humanoid | humanoid-standup |
|------------------------------------------------|----------|----------|----------|------------------|
| Baseline                                       | 4286     | 2527     | 6182     | 33093            |
| + LR Schedule                                  | **7154** | **3400** | 8366     | 33810            |
| + LR Schedule + LayerNorm                      | **7336** | 2574     | **8473** | **45793**        |
| + LR Schedule + Layer Norm + Weight Projection | **7186** | 2232     | **8677** | **50563**        |
| + LR Schedule + Weight Projection              | **6843** | 1276     | 7315     | 33982            |
| + LayerNorm                                    | 2764     | 2652     | 7080     | 39224            |
| + LayerNorm + Weight Projection                | 3379     | 2458     | 6911     | 31367            |
| + Weight Projection                            | 4629     | 922      | 4470     | 34256            |
|                                                |          |          |          |                  |

---

### Comment · Area_Chair_xik2 · 2024-08-07

Dear Reviewers,

The author responses have been uploaded. Please carefully review these responses to see if your concerns have been adequately addressed and actively participate in discussions.

It is important that **all reviewers should acknowledge having read the author responses by posting a comment**, irrespective of whether there is any change in your rating.

Thank you for your cooperation.

Best regards, \
Area Chair

---

### Decision · Program_Chairs · 2024-09-25

**Decision:**

Accept (poster)

**Comment:**

Adoptation of layer normalization implies that the growth of parameter norm will incur the decay of the effective learning rate, which will have detrimental effects on picking up new knowledge in the later stages of the training process. The authors solves this problem by periodically renormalizing the weights. The reviewers appreciated the simpleness and novelty of the proposed algorithm, as well as the robustness of the experiments. Although multiple concerns, including ones on the efficacy of NaP outside of Atari domains and others on the ambiguities in the experimental procedures, were raised, the authors have adequately addressed them during the discussion period. It is recommended to accept the paper.